

# Analysis of a saline dust storm from the Aralkum Desert - Part 1: Consistency of multisensor satellite aerosol products

Xin Xi[1], Jun Wang[2], Zhendong Lu[2], Andrew Sayer[3,4], Jaehwa Lee[3,5], Robert Levy[3], Yujie Wang[3,4], Alexei Lyapustin[3], Hongqing Liu[6,7], Istvan Laszlo[6,8], Changwoo Ahn[3,9], Omar Torres[3], Sabur Abdullaev[10], and Ralph Kahn[11]

[1]Department of Geological and Mining Engineering and Sciences, Michigan Technological University, Houghton, MI, USA
[2]Department of Chemical and Biochemical Engineering, The University of Iowa, Iowa City, IA, USA
[3]NASA Goddard Space Flight Center, Greenbelt, MD, USA
[4]Goddard Earth Sciences Technology and Research (GESTAR) II, University of Maryland, Baltimore County, Baltimore, MD, USA
[5]Earth System Science Interdisciplinary Center, University of Maryland, College Park, MD, USA
[6]Center for Satellite Applications and Research, National Environmental Satellite, Data, and Information Service, National Oceanic and Atmospheric Administration, College Park, Maryland, USA
[7]I. M. Systems Group, Inc., College Park, Maryland, USA
[8]Department of Atmospheric and Oceanic Sciences, University of Maryland, College Park, Maryland, USA
[9]Science Systems and Applications Inc., Lanham, MD, USA
[10]Physical Technical Institute of the Academy of Sciences of Tajikistan, Dushanbe, Tajikistan
[11]Laboratory for Atmospheric and Space Physics, The University of Colorado Boulder

**Correspondence:** Xin Xi (xinxi@mtu.edu)

**Abstract.** The performance and consistency of satellite observations in characterizing the saline dust emission from the newly formed Aralkum Desert have remained poorly understood. We address this knowledge gap by providing a review of satellite techniques capable of detecting the presence, column burden, and vertical height of airborne dust over desert surfaces. Then we evaluate the consistency between different aerosol products in observing an intense Aralkum dust storm in 2018, via syner-

gistic analyses of the ultraviolet aerosol index (UVAI) from OMPS, TROPOMI and EPIC, aerosol optical depth (AOD) from MODIS and VIIRS, and aerosol optical centroid height (AOCH) from CALIOP and EPIC. The UVAI products consistently delineate the areal extent of the freshly emitted dust plume if the dynamic range of each product is considered. The heavy dust plume is however erroneously masked as clouds in the AOD products. All UVAI products show large positive values over the Garabogazköl gulf and northern Caspian Sea due to enhanced UV absorption by turbid and saline waters, suggesting that cau-

tion must be taken to avoid misinterpreting the surface effect as dust signal over ephemeral or dried lakes. The AOD products show generally good agreement in observing the total and coarse-mode AOD associated with the dust outflow to Caspian Sea. Over-land AOD retrievals show strong non-linear relationships between aerosol algorithms. The NOAA Enterprise Processing System (EPS) product yields significantly lower AOD than other algorithms, likely due to the misuse of an urban aerosol optical model for dust retrieval. The EPIC AOCH retrieval shows the best agreement with CALIOP over heavy dust burden areas,

with both mean bias and RMSE below 0.5 km. This study reveals significant inconsistency between satellite aerosol products and the potential of multi-sensor approaches for identifying the product biases and limitations in Central Asia.





## 1 Introduction

Following the drying of Aral Sea over the past several decades, the newly formed Aralkum Desert has emerged as a major
source of wind-blown saline dust aerosol, causing a range of adverse impacts on the biodiversity, agriculture, and human well-
being across Central Asia (Orlovsky and Orlovsky, 2001; Abuduwaili et al., 2010; Xi and Sokolik, 2016). Aralkum stands
out among the vast drylands of Central Asia, not only due to its anthropogenic origin but also for the distinct chemical and
mineralogical composition of the erodible sediments. The Earth Surface Mineral Dust Source Investigation (EMIT) instrument
reveals that Aralkum contains more abundant carbonate and sulfate minerals, but less iron oxides and clays than the adjacent
sandy deserts (such as Karakum and Taklamakan deserts) (Fig. 1). Past studies confirmed that dust samples collected near
Aralkum contained significantly higher sulfate and chloride content than those near sandy deserts (Groll et al., 2013, 2019).
Hence, the dust aerosol originating from Aralkum Desert is expected to be more hygroscopic and less light-absorbing compared
to typical desert dust (Rudich et al., 2002). Even within the Aralkum Desert, there is substantial intra-basin variability in the
physiochemical properties of the dust-producing sediments. As the Aral Sea continued shrinking, the sediment grain size
became progressively smaller towards the lowest point of the basin, while evaporate minerals started to precipitate following
a typical sequence of calcite, gypsum/anhydrite, halite, and finally potassium and magnesium minerals. Consequently, distinct
spatial gradients in the mineralogical composition and abundance, grain size, and soil texture have been observed over the
Aral basin (Indoitu et al., 2015; Singer et al., 2001, 2003; Jiang et al., 2021). For example, Argaman et al. (2006) found
that the highly erodible takyr soils dominate the outer rim of Aralkum, whereas the newly formed solonchak soils are more
likely protected by salt crusts forming stable, coarse aggregates. Jiang et al. (2021) reported an increasing abundance of clay
and evaporites minerals and decreasing abundance of carbonates and organic carbon from the older coasts towards the newly
exposed seabed.

Quantifying the environmental impact associated with aeolian dust from Aralkum is greatly hampered by the lack of in situ
measurements of the spatially heterogeneous physical and chemical properties of the erodible sediments (e.g., soil texture,
mineralogical composition, crusting), and of the airborne dust particles (e.g., particle size distribution, shape, nonsphericity,
chemical composition or refractive index, mixing state, solubility). In particular, the global ground network of sun/sky pho-
tometers, AERosol RObotic NETwork (AERONET), has no operating sites near the Aralkum Desert, presumably because of
the region's remoteness and uninhabitable condition (Holben et al., 1998). The closest AERONET site is located 1000 km away
in the capital city of Tajikistan (Dushanbe) in a mountainous region with persistent urban pollution (Rupakheti et al., 2020).
AERONET measurements serve two important purposes. First, AERONET provides the climatological aerosol information
needed to specify the season- and region-dependent aerosol optical models in satellite aerosol algorithms (Dubovik et al.,
2002). Thus, the prevailing aerosol properties over Aralkum may be insufficiently represented in aerosol retrieval algorithms.
Second, AERONET measurements are used as ground truth for evaluating the regional biases of satellite aerosol retrievals,
which in turn help identify algorithm deficiencies and areas that need improvement. In general, satellite algorithms are op-
timized for global performance, but may be subject to significant local biases, if the local aerosol microphysical properties
deviate substantially from the algorithm assumptions.





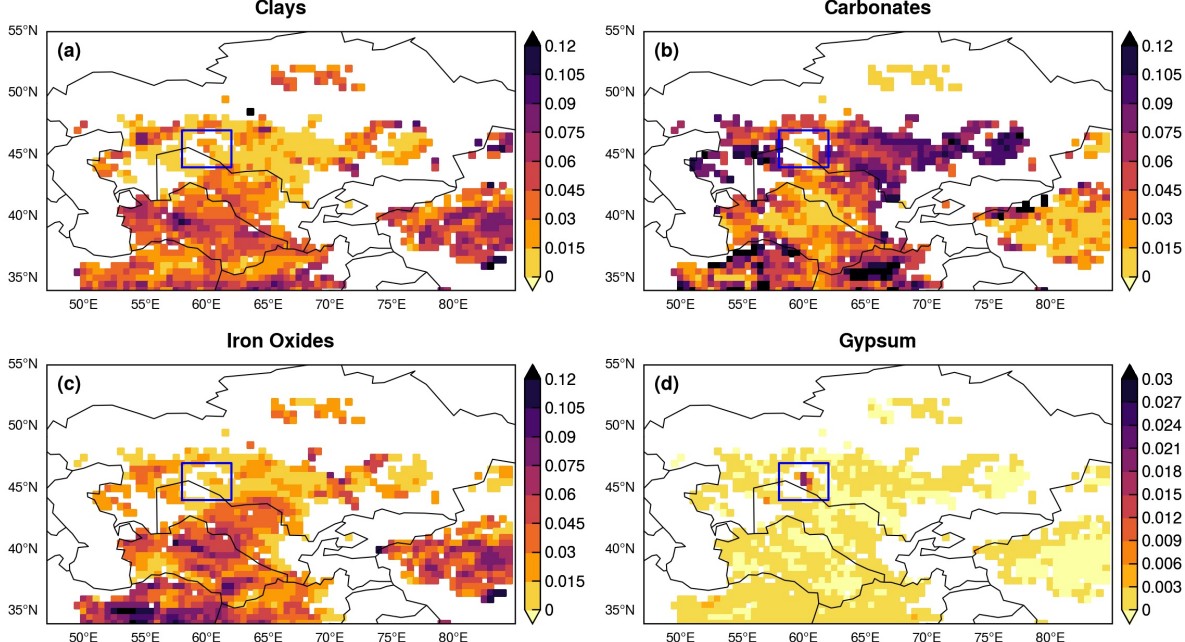

**Figure 1.** Aggregated spectral abundance of four mineral groups based on the Earth Surface Mineral Dust Source Investigation (EMIT) L3 product: (a) Clays (including chlorite, illite, muscovite, kaolinite, montmorillonite and vermiculite); (b) Carbonates (calcite and dolomite); (c) Iron Oxides (goethite and hematite); and (d) Gypsum. The boxed region indicates the Aralkum Desert. Note that panel (d) uses a different color scale from others.

Characterization of large-scale dust variability has primarily relied on spaceborne observations over the major continental outflow regions, such as the tropical Atlantic, North Pacific, and Arabian Sea (Yu et al., 2020). For mid-latitude drylands located in the interior of continents (such as Central Asia), observing airborne dust from space is a challenging task due to the difficulty of separating the influence of land surfaces from the top-of-the-atmosphere (TOA) radiance measurements. Central

Asia comprises a diversity of dust sources, including sandy and hilly deserts, steppes, salt flats, and ephemeral water bodies (Xi and Sokolik, 2015a, b). The heterogeneous and bright surfaces pose a major challenge for retrieving the aerosol properties in the visible and near-infrared (NIR) wavelengths. A number of satellite techniques and products have been developed to facilitate aerosol monitoring over desert areas. Data users may struggle with the product choice, not knowing the strengths and limitations of different products when applied to their region of interest. In this regard, a multisensor approach is preferred

over the use of a single product, especially for desert areas where aerosol remote sensing is subject to greater uncertainties than over remote oceans. So far, there is poor understanding of the performance and consistency of satellite aerosol products in observing the saline dust from the Aralkum Desert.

During 27–29 May 2018, an intense dust storm occurred from the Aralkum Desert due to cold air outbreak from the polar latitudes, causing persistent haze, record high particulate concentration, and salt deposition on cotton and food crops (Xi, 2023).



This event was described as "raining salt" in the media, due to the high salt content in the lofted particles (Radio Free Europe
Radio Liberty, 2018). This paper documents the first part of a detailed analysis of the saline dust storm, and investigates the
consistency of multi-sensor satellite aerosol products in detecting the presence, column burden, and vertical height of airborne
dust from the Aralkum Desert. A companion paper will focus on the atmospheric dynamics and model simulations of the event.
In this paper, we first provide a review of current satellite remote sensing techniques with aerosol observation capability over

desert regions (such as Central Asia) (Section 2). Specifically, the review focuses on the physical principles and algorithm
treatment of aerosol and land surface properties associated with the retrieval of three aerosol parameters: ultraviolet aerosol
index (UVAI), mid-visible aerosol optical depth (AOD), and aerosol optical centroid height (AOCH). The review provides
insight into the inherent differences between different aerosol algorithms, including the algorithms designed to retrieve the
same aerosol parameter from the same sensor. Then, we perform an intercomparison of multiple UVAI, AOD, and AOCH

products to examine the cross-sensor and cross-algorithm consistency in observing the Aralkum dust storm of May 2018
(Section 3). Due to the lack of ground truth, we are unable to evaluate the accuracy of the AOD products. Instead, we focus on
the consistency (or lack thereof) between different sensors, platforms, and algorithms, and explore the synergistic use of multi-
sensor data to identify possible limitations and biases in the aerosol products. The major findings of the study are summarized
in Section 4.

## 2 Review of Aerosol Remote Sensing Over Desert Areas

### 2.1 UVAI

#### 2.1.1 Overview

In the near UV wavelengths (330–380 nm) where ozone absorption is weak, the TOA upwelling radiance consists of contribu-
tions from molecular scattering, scattering and absorption by clouds and aerosols, and surface reflection. The low reflectivity

of snow-free land surfaces in the UV allows the separation of the atmospheric contribution (called path radiance) from sur-
face reflection (Herman and Celarier, 1997). The aerosol detection capability in the near UV spectrum was first discovered
as a spectral residual quantity in the Total Ozone Mapping Spectrometer (TOMS) ozone algorithm, which assumes that the
aerosol and cloud scattering and surface reflection can be collectively represented as a hypothetical Lambertian reflector at the
bottom of a pure molecular atmosphere. The Lambert Equivalent Reflectivity (LER) of this "effective surface" is assumed to

be wavelength-independent, and determined from radiance measurements at a reference wavelength ($\lambda_0$) insensitive to ozone
absorption, based on radiative transfer calculations for a pure molecular atmosphere (Herman and Celarier, 1997). A resid-
ual quantity in the LER (or LER difference) between $\lambda_0$ and a neighboring wavelength ($\lambda$, usually shorter than $\lambda_0$), which
measures the deviation of the spectral dependence of observed UV radiances from that of a pure molecular atmosphere (gov-
erned by Rayleigh scattering), was found to be close to zero under most conditions. Significant LER differences between $\lambda_0$

and $\lambda$, however, were frequently found to coincide with the presence of UV-absorbing aerosols, such as dust, carbonaceous



aerosol, and volcanic ash (Hsu, 1996; Herman et al., 1997). Because of the predominant association with absorbing aerosols, the spectral residual quantity has been referred to as UVAI, or Absorption Aerosol Index (AAI).

The computation of UVAI involves minimal a priori information about aerosol properties, but highly depends on the calibration accuracy at the two wavelengths involved. Historically, two wavelength pairs have been used: 340/380 nm and 354/388 nm, of which the longer wavelength is used as the reference wavelength $\lambda_0$. In the original UVAI definition, clouds were treated as part of the Lambertian reflector in Rayleigh scattering calculations. This LER-based UVAI yields small negative values for clouds and non-absorbing aerosols. Torres et al. (2018) introduced a new approach by explicitly treating the scattering effects of water clouds based on Mie theory. The Mie-based UVAI was found to be closer to zero over cloudy scenes, and have weaker angular dependence compared to the LER-based UVAI.

UVAI yields large positive values in the presence of absorbing aerosols, including over deserts, snow, or ice-covered areas, and above clouds (Torres et al., 2012). This makes UVAI an excellent tracer of airborne dust. For example, the TOMS UVAI product was used for tracking dust transport and mapping dust sources, which established the basis for the empirical dust source function used in global dust models (Prospero et al., 2002). Compared to a pure Rayleigh scattering atmosphere, dust reduces the spectral contrast of backscattered UV radiances through the absorption of Rayleigh scattered radiation from beneath the aerosol layer (Herman et al., 1997; Torres et al., 1998; de Graaf et al., 2005). In addition, dust is generally more absorbing at shorter wavelengths, which further enhances the deviation of the spectral dependence of backscattered radiances from that of a Rayleigh scattering atmosphere (Sokolik and Toon, 1999). For an aerosol layer, its UVAI value is closely related to the AOD, vertical height, absorption capability (i.e. single scattering albedo) and surface albedo (de Graaf et al., 2005). The more elevated an aerosol layer, the higher the UVAI, holding other things equal. It also means that assumptions or a priori information about the aerosol height and type are required to estimate AOD from UVAI (Torres et al., 2013).

Despite the unique capability of UVAI in detecting absorbing aerosols over various surface types, it should be noted that some geophysical effects can cause large UVAI values similar to the effects of aerosols, such as sun glint, ocean color, and the wavelength dependence of surface reflectance that violates the spectrally independent LER assumption (Herman and Celarier, 1997). These effects, if misinterpreted as aerosol signal, will introduce errors in inferring dust properties and sources from UVAI.

UVAI products has been generated from a number of spaceborne UV-visible spectrometers, including TOMS, Global Ozone Monitoring Experiment (GOME), Scanning Imaging Absorption Spectrometer for Atmospheric Cartography(SCIAMACHY), Ozone Monitoring Instrument (OMI), Ozone Mapping and Profiler Suite (OMPS), and TROPOspheric Monitoring Instrument (TROPOMI). Below we briefly describe the instruments considered in this study.

### 2.1.2 OMI

OMI is a nadir-looking grating spectrometer that operates onboard the Aura spacecraft in a sun-synchronous orbit with an ascending node equatorial crossing time of 13:38. OMI measures upwelling radiances in three channels: UV-1 (264–311 nm), UV-2 (307–383 nm) and VIS (349–504 nm) with a footprint size of 13×24 km$^2$ at nadir. The OMI near-UV aerosol algorithm (OMAERUV) reports the Mie-based UVAI at the 354/388 nm wavelengths (Torres, 2006). Unfortunately, OMI suffered a row



anomaly issue since 2008 (Torres et al., 2018), which significantly reduced the data coverage and thus is not included in our
analysis.

### 2.1.3    OMPS

The OMPS Nadir Mapper is an imaging spectrometer that operates onboard the Suomi National Polar-orbiting Partnership
(SNPP) spacecraft in a sun-synchronous orbit with an ascending node equatorial crossing time of 13:30. OMPS measures UV

radiances every 0.42 nm from 300 to 380 nm with a nadir footprint size of $50{\times}50$ km$^2$ and $110°$ across-track field of view,
equivalent to a ground swath of 2800 km. The OMPS Nadir Mapper aerosol algorithm (NMMIEAI) reports the Mie-based
UVAI at the 340/378.5 nm wavelengths (Torres, 2019a).

### 2.1.4    TROPOMI

TROPOMI is a nadir-viewing, push-broom-type grating spectrometer deployed on the Copernicus Sentinel 5 Precursor (S5P)

mission, which flies in close formation with the SNPP spacecraft (less than 5 min apart) in a sun-synchronous orbit with an
ascending node equatorial crossing time of 13:30 (Veefkind et al., 2012). TROPOMI measures reflected and emitted radiation
in the UV-visible (270–500 nm), NIR (710–770 nm), and shortwave infrared (SWIR, 2314–2382 nm) wavelengths with a
footprint size of $3.5{\times}7$ km$^2$ at nadir.

There are two separate UVAI products from TROPOMI, one developed by Royal Netherlands Meteorological Institute

(KNMI) and European Space Agency (ESA), and the other developed by the National Aeronautics and Space Administration
(NASA). The ESA version reports LER-based UVAI at two wavelength pairs: 354/388 nm and 340/380 nm (Stein Zweers,
2022). The NASA version is based on the TropOMAER aerosol algorithm, and reports both LER- and Mie-based UVAI at the
354/388 nm wavelengths (Torres, 2021).

### 2.1.5    EPIC

EPIC is an imaging spectroradiometer onboard the Deep Space Climate Observatory (DSCOVR) spacecraft, and measures the
Earth's reflected solar radiance in ten narrow channels spanning from UV to NIR (317–779 nm), with a spatial resolution of 8
km at 443 nm and 16 km in other bands (Marshak et al., 2018). DSCOVR operates in a Lissajous orbit about Lagrange-1 point
in the Earth-Sun system, which allows EPIC to view the sunlit disk of Earth every 60–100 minutes. The EPIC EPICAERUV
aerosol algorithm reports both LER- and Mie-based UVAI at the 340/388 nm wavelengths (Torres, 2019b).

## 2.2    AOD

### 2.2.1    Overview

Unlike UVAI which is a qualitative tracer of absorbing aerosols, mid-visible (usually at 0.55 $\mu$m) AOD provides a quantitative
measure of the aerosol burden in the atmospheric column. Inferring AOD from reflected sunlight by the surface-atmosphere
system is an ill-posed inverse problem and must rely on a priori knowledge or assumptions on the aerosol microphysical



properties, known as aerosol optical models. The aerosol optical models are characterized by unique combinations of particle size distribution, shape, and composition (or refractive index). A general strategy of AOD retrieval from passive, downward-looking sensors involves using a forward radiative transfer model to compute the TOA spectral reflectances for a set of pre-defined aerosol optical models, surface types, and solar/viewing geometries expected to be encountered by the instruments (Remer et al., 2013a). During the AOD retrieval, the pre-calculated reflectances, which are stored in look-up tables (LUTs),

are compared against measured TOA reflectances to find the best fit and solution for AOD and additional parameters related to particle size and absorption. Aerosol retrieval using this method is highly sensitive to the predefined aerosol models, which aim to represent the aerosol conditions encountered by the sensor as realistically as possible, and ultimately determine the AOD spectral dependence (i.e. Ångström Exponent) and single scattering albedo (SSA). The predefined aerosol models are commonly determined from long-term sun/sky photometer measurements and/or field campaign measurements, and typically

represent the prevailing aerosol conditions at different locations and seasons (Dubovik et al., 2002). However, different aerosol algorithms may include inconsistent, sometimes conflicting, assumptions of the aerosol properties, causing disparities between satellite products and between satellite observations and aerosol models.

Below, we briefly review the AOD algorithms implemented in the Moderate Resolution Imaging Spectroradiometer (MODIS) and Visible Infrared Imaging Radiometer Suite (VIIRS) sensors. The Dark Target (DT), Multi-Angle Implementation of Atmo-

spheric Correction (MAIAC), and Deep Blue (DB) algorithms are suitable for observing dust outflow over major water bodies, such as the Caspian Sea. The DB, MAIAC, and Enterprise Processing System (EPS) algorithms are capable of retrieving AOD over desert surfaces. The review focuses on the treatment of aerosol optical models and spectral surface reflectances in different algorithms. Other algorithm processes, such as the screening of sun glint/snow/clouds, are not discussed. A detailed review of the treatment of dust microphysical properties in satellite algorithms and global models was recently given by Castellanos

et al. (2024).

### 2.2.2  DT algorithm

The DT algorithm exploits the contrast of reflective aerosol layers against a dark background in the visible spectrum. Over ocean, the algorithm searches the LUT for the best fit of the measured TOA reflectances at seven window bands from visible to SWIR, and obtains solutions for both AOD and fine mode fraction (FMF) at 0.55 $\mu$m (Tanré et al., 1997). The algorithm

assumes that the ambient aerosol scene can be represented as a linear combination of one fine mode and one coarse mode weighted by FMF, out of 20 possible combinations of four fine-mode and five coarse-mode aerosol models (Remer et al., 2005). The aerosol models are derived from long-term AERONET measurements near water bodies. Two of the five coarse models represent dust aerosol, with the refractive index and size distribution based on observed airborne dust. Due to lack of non-spherical dust models, the algorithm was known to generate biased retrievals over dusty oceanic regions (Zhou et al.,

2020b). Recently, a spheroidal dust model was introduced to improve retrieval for dusty conditions (Zhou et al., 2020a).

Over land, the DT algorithm uses three bands (0.47, 0.65, and 2.11 $\mu$m for MODIS; 0.49, 0.67, and 2.25 $\mu$m for VIIRS) in which the surface reflectivity of vegetated lands is still low (Kaufman et al., 1997a). The algorithm assumes a number of fine- and coarse-dominated aerosol models, which are prescribed as a function of season and location based on cluster analysis of




the AERONET climatology (Levy et al., 2013). The algorithm accounts for the surface contribution to the TOA reflectances

based on empirical relationships between the visible and SWIR (2.11 or 2.25 $\mu$m) surface reflectances, which are applicable
for dark surfaces only (Kaufman et al., 1997b; Remer et al., 2005). Because the assumed visible-to-SWIR reflectance ratios do
not apply for bright desert areas, the DT algorithm provides very limited coverage near dust source areas.

### 2.2.3   DB algorithm

Inspired by the aerosol observation capability in the near-UV, the DB algorithm employs the 0.41 $\mu$m or "deep blue" band,

which has lower and more homogeneous surface reflectivity than the longer visible wavelengths and thus provides sufficient
contrast between the surface and aerosol signals (Hsu et al., 2004). Over land, the algorithm uses three bands: deep blue
(0.41 $\mu$m), blue (0.47 $\mu$m for MODIS, 0.49 $\mu$m for VIIRS), and red (0.65 $\mu$m for MODIS, 0.67 $\mu$m for VIIRS). Initially
implemented to fill the gap over bright surfaces (including deserts), the algorithm considered an aerosol optical model with
an empirical scattering phase function that matched the AERONET measurements at Cape Verde. These were later replaced

with a spheroidal dust model and a spherical fine-dominated anthropogenic model, while the algorithm is expanded to the
entire land surfaces not influenced by cloud, snow, or ice. The dust/anthropogenic optical models are created for various SSA
values, which are prescribed for specific locations and seasons. Note that the MODIS DB product used in this study dose not
include the aerosol optical model update (Hsu et al., 2013), while the VIIRS DB product does (Lee et al., 2024). The surface
reflectance is determined based on a pre-calculated database over bright surfaces (e.g., deserts, urban areas), and empirical

surface reflectance relationships between the visible and SWIR bands (similar to the DT approach) as a function of normalized
difference vegetation index (NDVI) over vegetated areas (Hsu et al., 2013).

Over ocean, the DB algorithm began to generate AOD retrieval from VIIRS based on the Satellite Ocean Aerosol Retrieval
(SOAR) algorithm (Sayer et al., 2018). SOAR considers four aerosol optical models over water: maritime, dust, fine-dominated,
and mixed, each represented as a bi-modal distribution of one fine and one coarse modes. The dust model consists of one

spheroidal fine mode and one spheroidal coarse mode, which currently is derived from AERONET measurements at Cape Verde
(Lee et al., 2017).

### 2.2.4   EPS algorithm

NOAA has developed a suite of aerosol products from the new-generation polar-orbiting (e.g., VIIRS) and geostationary (e.g.,
Advanced Baseline Imager) sensors using the EPS algorithm. Here we focus on the VIIRS EPS AOD product, described in

Laszlo (2018) and more recently Laszlo and Liu (2022).

Over ocean and inland water, the EPS algorithm is based on the MODIS heritage. The aerosol column is represented as a
linear combination of one spherical fine and one spherical coarse mode, selected from four fine mode and five coarse mode
candidate models, weighted by varying fractions of the fine mode (FMF). The aerosol models are based on the same models
considered in the MODIS DT algorithm (Remer et al., 2006). The algorithm searches for the AOD and FMF that give the

best match between observed and pre-calculated TOA reflectances at seven VIIRS channels (M4, M5, M6, M7, M8, M10, and





M11) (Jackson et al., 2013; Laszlo and Liu, 2022). The surface contribution to the TOA reflectance is represented as a sum of bi-directional sun-glint and Lambertian underwater and whitecap reflection.

Over land, the EPS algorithm simultaneously retrieves AOD, aerosol model, and Lambertian surface reflectances in selected bands, by matching observed and calculated TOA reflectance over both dark and bright (snow-free) surfaces. Over dark land, the surface reflectance is retrieved from either the observed M5 (0.672 $\mu$m) or M11 (2.25 $\mu$m) band TOA reflectance, followed by estimation of surface reflectances in other bands using predetermined relationships between surface reflectances in bands M1, M2, M3, M5 and M11. The relationships are functions of observation geometry and surface type. Over bright lands, the surface reflectance is estimated from a static database of M1/M5, M2/M5 and M3/M5 reflectance ratios that are functions of the scattering angle (Zhang et al., 2016). During the retrieval process, the EPS algorithm tries four candidate models, of which three are fine-mode dominated aerosols with spherical shapes (labeled as generic, urban, and smoke) and one is coarse-mode dominated aerosol with non-spherical shape (labeled as dust). The exception is the surface region of North Africa and the Arabian Peninsula where the algorithm uses only the dust model since that is the most dominant aerosol type and because the dynamic model selection is impaired by the bright surface. The candidate aerosol models are adopted from the Collection 5 MODIS DT algorithm (Remer et al., 2006; Levy et al., 2007).

### 2.2.5 MAIAC algorithm

The MAIAC algorithm uses the dynamic minimum reflectance method to define the surface reflectance spectral ratios for each 1 km grid cell, which allows AOD retrievals over dark and bright surfaces at a high spatial resolution (1×1 km$^2$). MAIAC also uses a sliding window technique to accumulate up to 16 days of multi-angle measurements from different orbits for the same location to retrieve bidirectional surface reflectance over land (including deserts) (Lyapustin et al., 2018). The C6.1 MAIAC uses nine aerosol optical models derived from AERONET data climatology analysis to represent the regional background aerosol conditions. For Central Asia, the background aerosol model ("Model 2") is derived from AERONET measurements over western USA, and represents a mixture of dust and fine mode aerosols. During the retrieval, a smoke/dust test is applied to determine whether the background or dust model should be used in the retrieval. Once dust is detected, the algorithm uses a spheroidal dust model ("Model 6") derived from AERONET measurements from the Solar Village site in Saudi Arabia (Dubovik et al., 2002, 2006).

### 2.2.6 Algorithm implementation in MODIS and VIIRS

Under NASA's Earth Observing System program, the twin MODIS sensors onboard the Terra and Aqua spacecrafts have generated more than 20 years' global aerosol records. Terra operates on a sun-synchronous orbit with a 10:30 descending node equatorial crossing time. Aqua has an ascending node equatorial crossing time of 13:30 and provides nearly coincident observations with other sensors as part of the Afternoon-train (or A-train) constellation. In the MODIS aerosol product (designated as MOD04 for Terra and MYD04 for Aqua), the DT and DB algorithms perform retrieval at a nominal resolution of 10×10 km$^2$, providing data coverage over dark and bright surfaces, respectively (Levy et al., 2013). In the Collection 6 release, a finer resolution DT product was produced at a 3×3 km$^2$ resolution (referred to as DT3K) to better resolve small-scale aerosol





features (Munchak et al., 2013; Remer et al., 2013b). The DT3K product uses the same DT technique, but with different pixel
aggregation and quality assurance (QA) rules. In addition, the MAIAC algorithm uses data from both Terra and Aqua to conduct aerosol retrieval at a high resolution of 1×1 km$^2$ over both water and land (designated as MCD19A2) (Lyapustin et al., 2018).

As the successor for MODIS, VIIRS currently operates on three Joint Polar Satellite System (JPSS) satellites: SNPP, NOAA-20, and NOAA-21, which fly in sun-synchronous orbits with an ascending node equatorial crossing time around 13:30. The DT algorithm has been ported to VIIRS, and performs AOD retrieval at a nominal resolution of 6×6 km$^2$ (Sawyer et al., 2020). The DB algorithm implementation in VIIRS is different from that for MODIS, and performs AOD retrieval both over ocean and over land (Hsu et al., 2019; Lee et al., 2024). Hence, the VIIRS DB AOD product is provided as an alternative to the VIIRS DT AOD product. In addition to the DB and DT products from NASA, the NOAA EPS AOD product is another alternative for the VIIRS sensors, and performs retrieval at the pixel level with a resolution of 0.75×0.75 km$^2$ (Laszlo and Liu, 2022).

## 2.3  AOCH

### 2.3.1  Overview

Aerosol vertical distribution affects the atmospheric lifetime and impact of aerosols, and is an essential input for various applications, such as retrieving AOD from UVAI and estimating ambient PM2.5 concentration from AOD (Li et al., 2015; Sun et al., 2019). The vertical height of absorbing aerosols (such as dust) is of particular interest for weather and climate research, due to their radiative impact on the atmospheric thermodynamics and stability (Samset et al., 2018). Currently, the retrieval of aerosol vertical distribution is obtained from both active and passive sensors. Active sensors, such as the spaceborne lidars onboard the Cloud-Aerosol Lidar and Infrared Pathfinder Satellite Observations (CALIPSO) and Cloud-Aerosol Transport System (CATS), provide vertically-resolved measurements of aerosol volume extinction at high vertical resolutions (Winker et al., 2009; Yorks et al., 2016). An important limitation of spaceborne lidars, however, is their narrow swath and poor spatial coverage. In recent years, there has been substantial progress in retrieving one piece information of aerosol vertical distribution from passive sensors based on various techniques, including stereoscopic retrieval from polar-orbiting multiangle or geostationary imagers (Nelson et al., 2013; Carr et al., 2020), polarimetric observations in the near UV (Wu et al., 2016), differential optical absorption spectroscopy (DOAS) in the NIR (e.g., oxygen A and B bands) (Xu et al., 2017, 2019; Chen et al., 2021), hyperspectral thermal infrared (TIR) measurements (Pierangelo et al., 2004; Peyridieu et al., 2010), and the retrieval of smoke injection height from the thermal contrast of elevated smoke aerosol (Lyapustin et al., 2020). Although passive techniques do not achieve the same level of accuracy as lidars and only estimate an effective height with limited information on the aerosol layer geometric thickness, they provide much better spatial coverage and revisit frequency (Lu et al., 2021, 2023).

The definition of aerosol layer height vary by different techniques, and may refer to the top of aerosol layers such as from stereoscopic techniques, or an effective central height corresponding to peak aerosol extinction (Xu et al., 2017; Yorks et al., 2023). Similar to AOD, the aerosol layer height is an optical quantity, and expected to vary with the wavelength at which they





are retrieved. Radiation at longer wavelengths is generally capable of penetrating deeper into the aerosol layer than radiation at shorter wavelengths. In addition, except for stereoscopic techniques, passive retrieval of aerosol height depends on the aerosol vertical distribution, and must rely on assumptions about the aerosol layer thickness or profile. Below, we describe two aerosol layer height products with coverage over Central Asia, including the aerosol extinction-weighted height derived from CALIPSO, and the EPIC AOCH product.

### 2.3.2 CALIOP aerosol extinction-weighted height

As part of the A-train constellation, CALIPSO provides critical information on the vertical distribution and optical properties of aerosols and clouds. CALIPSO carries an elastic backscatter lidar, Cloud-Aerosol Lidar with Orthogonal Polarization (CALIOP), which measures polarized backscatter at 532 nm and total attenuated backscatter at 532 nm and 1032 nm, with a vertical resolution of 30 m below an altitude of 8.2 km and 60 m between 8.2 and 20.2 km (Winker et al., 2009). In the CALIOP level-2 data processing, the calibrated attenuated backscatter coefficient profiles are used to detect the top and base altitudes of atmospheric features using a selective iterated boundary location (SIBYL) algorithm. Then a set of scene classification algorithms (SCA) are used to classify these features as aerosol and cloud, and determine the aerosol type and cloud phase. During this process, the aerosol extinction-to-backscatter ratios (or aerosol lidar ratios) are selected to derive the aerosol extinction and backscatter coefficient profiles (Young et al., 2018). The lidar ratio selection relies on the aerosol typing algorithm which considers six tropospheric aerosol types: clean marine, dusty marine, dust, polluted continental/smoke, polluted dust, and elevated smoke, each assigned a default lidar ratio (Omar et al., 2009; Kim et al., 2018). The aerosol type is determined based on several input parameters: altitude, location, surface type, particulate linear depolarization ratio, and integrated attenuated backscatter. Specifically, thresholds of layer-integrated volume depolarization ratios are used to detect non-spherical aerosol types, including pure dust and dust mixtures with marine aerosol ("dusty marine") or polluted continental aerosol ("polluted dust") (see Fig. 2 in Omar et al., 2009). Based on the aerosol extinction profile, the CALIOP AOCH with respect to the mean sea level can be computed as $\frac{\sum_{i=1}^n \beta_{ext,i} Z_i}{\sum_{i=1}^n \beta_{ext,i}}$, where $\beta_{ext,i}$ is the 532 or 1064 nm aerosol extinction coefficient (km$^{-1}$) at level $i$, and $Z_i$ is the altitude (km) at level $i$ (Koffi et al., 2012).

### 2.3.3 EPIC AOCH product

The AOCH retrieval from EPIC employs the oxygen absorption spectroscopic measurements in the A band near 760 nm and B band near 688 nm. The physical principle is that the enhanced scattering from elevated aerosol layers shortens the photon path length of reflected sunlight, thereby reducing its chance of being absorbed by oxygen molecules (Kokhanovsky et al., 2015; Ding et al., 2016). The oxygen A and B bands are characterized by different penetration depths of solar radiation and altitude-dependence of oxygen absorption, which establishes a basis for the AOCH retrieval from the enhanced TOA reflectances compared to a pure (aerosol-free) Rayleigh scattering atmosphere (Xu et al., 2019).

The AOCH retrieval algorithm employs a set of LUTs consisting of pre-computed TOA reflectances for a range of AOD, AOCH, surface reflectivity, and surface pressure values for the solar/viewing geometries expected to be encountered by EPIC. Six EPIC bands are considered, including the oxygen A and B bands at 764 and 688 nm and two reference continuum bands at





780 and 680 nm. The aerosol vertical distribution is represented by a quasi-Gaussian profile characterized by a centroid altitude and a fixed half-width at half maxima of 1 km. The centroid altitude is reported as the aerosol layer height in the AOCH product, and represents the altitude at which the aerosol volume extinction peaks. Currently, the EPIC AOCH product considers three aerosol optical models: smoke, Saharan dust, and Asian dust. The dust models are derived from AERONET measurements at Cape Verde and over East Asia, respectively (Xu et al., 2017). The surface reflectivity is represented using different approaches

for water and land. The water reflectivity is based on the GOME-2 surface LER database (Xu et al., 2017), whilst the land surface reflectivity is derived from the MODIS bidirectional reflectance distribution function (BRDF) and albedo products (Xu et al., 2019). During the retrieval, the AOD is first retrieved at the EPIC window channel of 443 nm. Then the observed absorption-to-continuum band reflectance ratios or DOAS ratios, which are defined as the ratio of reflectances in the oxygen absorption bands to the reflectances in the reference continuum bands, are matched with the LUTs to find the best solutions

for AOCH using different spectral fitting strategies over water and land. In general, the sensitivity of DOAS ratios to AOCH is the highest for heavy aerosol burden and low surface reflectivity conditions, under which the retrieved AOCH is considered the most accurate. Whereas, the retrieved AOCH becomes less reliable when AOD decreases to nearly zero and/or surface reflection dominates the aerosol signal in the TOA reflectances.

## 3    Evaluation of Cross-sensor and Cross-algorithm Consistency

Table 1 summarizes the satellite aerosol products and parameters considered in this study. The products are selected based on data availability and are mostly from polar-orbiting sensors (except EPIC) with similar overpass time over Aralkum Desert. For the AOD products, we use the QA-screened, best estimate AOD parameters when available; otherwise we apply QA screening as recommended by each product. While some of the considered products have duplicated parameters (e.g., AOD is available from the NASA UVAI products), we focus on the primary and most widely used aerosol parameter from each product. The

UVAI and AOD products also report parameters related to aerosol size (e.g., Angström Exponent) and absorption (i.e. single scattering albedo); however, they are subject to larger biases and usually not recommended for use in scientific studies.

### 3.1    Horizontal and vertical dust distributions

In this section we describe the dust plume trajectory based on nearly coincident observations from passive and active sensors. Fig. 2 shows the daytime MODIS/Aqua true color images on 27 and 29 May 2018 and the nighttime Spinning Enhanced

Visible and InfraRed Imager (SEVIRI) Dust RGB composite image on 28 May 2018, along with the 532 nm total attenuated backscatter profile and AOD from three coincident CALIOP overpasses. On 27 May 2018, MODIS observed an extensive whitish dust plume off the Aralkum Desert and advancing towards Iran and Afghanistan. CALIOP missed the Aralkum dust plume, but detected a shallow dust layer (0–1 km) at the southeast coast of Caspian Sea. The dust layer was likely generated from the dry channel of ancient River Uzboy in western Turkmenistan, which has been identified as one of the most active dust

sources in Central Asia (Nobakht et al., 2021).





**Table 1.** List of satellite aerosol products considered in this study.

| Sensor/Platform | Product | Parameter | Resolution | Reference |
|---|---|---|---|---|
| **UV Aerosol Index (UVAI)** | | | | |
| OMPS/SNPP | NMMIEAI | UVAerosolIndex | $50 \times 50$ km$^2$ | Torres (2019a) |
| TROPOMI/S5P | TROPOMAER | UVAerosolIndex | $7.5 \times 3$ km$^2$ | Torres (2021) |
| EPIC/DSCOVR | AER | UVAerosolIndex | $12 \times 12$ km$^2$ | Torres (2019b) |
| TROPOMI/S5P | AER | aerosol_index_340_380 aerosol_index_354_388 | $7.5 \times 3$ km$^2$ | ESA (2021) |
| **Aerosol Optical Depth (AOD)** | | | | |
| MODIS/Aqua | DT v6.1 | Optical_Depth_Land_And_Ocean | $10 \times 10$ km$^2$ | Levy and Hsu (2015a) |
| MODIS/Aqua | DT3K v6.1 | Optical_Depth_Land_And_Ocean | $3 \times 3$ km$^2$ | Levy and Hsu (2015b) |
| MODIS/Aqua | DB v6.1 | Deep_Blue_Aerosol_Optical_Depth _550_Land_Best_Estimate | $10 \times 10$ km$^2$ | Levy and Hsu (2015a) |
| MODIS/Aqua | MAIAC v6.1 | Optical_Depth_055 | $1 \times 1$ km$^2$ | Lyapustin and Wang (2022) |
| VIIRS/NOAA20 | DT v2 | Optical_Depth_Land_And_Ocean | $6 \times 6$ km$^2$ | Levy et al. (2023) |
| VIIRS/NOAA20 | DB v2 | Aerosol_Optical_Thickness _550_Land_Ocean_Best_Estimate | $6 \times 6$ km$^2$ | Hsu (2022) |
| VIIRS/NOAA20 | EPS v3r0 | AOD550 | $0.75 \times 0.75$ km$^2$ | Kondragunta et al. (2023) |
| **Aerosol Optical Centroid Height (AOCH)** | | | | |
| CALIOP/CALIPSO | 05kmAPro v4.51 | Extinction_Coefficient_532 Extinction_Coefficient_1064 | 5 km | ASDC (2023) |
| EPIC/DSCOVR | AOCH v1 | aod, height, surf_refl | $30 \times 30$ km$^2$ | ASDC (2018) |

On 28 May 2018, a high pressure system developed just to the south of Aralkum Desert, which transported the lofted dust across the Ustyurt Plateau to the Caspian Sea. The nighttime CALIOP overpass detected an extensive dust layer stretching from Aralkum to the Kopet-Dag mountain range at the Iran-Turkmenistan border. The Kopet-Dag Range acted as a physical barrier of the dust plume, causing dust particles to accumulate along the foothills resulting in high AODs up to 3. The dust

layer was elevated to an altitude of 1–2 km near Aralkum, but extended to the ground surface while approaching the gently sloping foothill region. The relatively low dust layer height was probably due to the prevailing high pressure system and shallow atmospheric boundary layer during nighttime. The varying dust layer height may have explained the different color contrasts shown in the SEVIRI image. Specifically, the elevated dust displays a rich magenta color and strong contrast near the Aralkum. Whereas, the dust layer near the foothills was less discernible, possibly due to reduced temperature contrast between

the ground surface and dust aloft.

On 29 May 2018, the remnants of the Aralkum dust storm continued to affect western Uzbekistan and Turkmenistan. The suspended dust was only partially detected by CALIOP due to the presence of extensive clouds above the dust. The dust layer





**Figure 2.** (a, b, c) CALIPSO ground tracks superimposed on nearly coincident MODIS/Aqua true color and SEVIRI Dust RGB composite images on 27–29 May 2018. The ground track start and end times are indicated. The MODIS or SEVIRI scan time over the Aralkum Desert (45°N, 60°E) is shown at the top right corner. (d, e, f) The corresponding CALIOP 532 nm total attenuated backscatter profile and AOD (magenta dots). Black lines indicate the ground surface.

reached an altitude of 3.5 km, likely due to convective mixing in the daytime boundary layer and enhanced vertical motion ahead of a cold front. The CALIOP-retrieved AOD was close to 2 in many areas.

## 3.2 Comparison of UVAI products

Fig. 3 compares five UVAI products in detecting the freshly emitted dust plume on 27 May 2018. The UVAI products include the Mie-based UVAI (named as "UVAerosolIndex") from OMPS, TROPOMI, and EPIC, provided by NASA, and the LER-based UVAI from TROPOMI, provided by ESA. These sensors flew over Aralkum within a short time window (less than 20





**Figure 3.** Comparison of five UVAI products on 27 May 2018: (a) MODIS/Aqua true color image, (b) 340/379 nm UVAI from OMPS, (c) 354/388 nm UVAI from TROPOMI (NASA), (d) 340/388 nm UVAI from EPIC, (e) 354/388 nm UVAI from TROPOMI (ESA), (f) 340/380 nm UVAI from TROPOMI (ESA), and (g) violin plots and summary statistics of the UVAI products. The instrument overpass time over the Aralkum Desert (45°N, 60°E) is shown at the top right corner of each panel. Straight lines in (a–f) indicate the CALIPSO ground track on 27 May 2018. Black contours in (b–f) indicate the 95% percentile of each product. Summary statistics in (g) include pixel count (N), mean, median, minimum, maximum, 95% percentile (Q95), and the fraction of pixels with UVAI≥2 (F2).



minutes) and thus observed the same scene. The products show similar spatial patterns associated with the dust plume, but

reveal significant differences in the UVAI magnitude and dynamic range.

By accounting for the cloud scattering effects, the Mie-based UVAI products from OMPS and TROPOMI (NASA) yield near zero values for dust-free regions. Both products display a bimodal distribution with a peak around zero, and a small positive peak close to 1 (Fig. 3g). The OMPS UVAI is generally smaller likely due to its coarser resolution and cloud influence. EPIC yields significantly higher UVAI than OMPS and TROPOMI (NASA) for dusty scenes, partly because of the higher sensitivity

to absorbing aerosols in the 340/380 nm wavelengths. The cloudy scenes in EPIC appear more noisy, most likely due to EPIC L1 calibration issues (personal communication with Karin Blank).

The LER-based UVAI products from TROPOMI (ESA) are lower than the NASA counterpart, and show large negative values over dust-free areas. The choice of wavelength pairs (354/388 vs. 340/380 nm) has a minor effect on the shape of the UVAI statistical distribution, both characterized by a tri-modal distribution with two negative modes and one positive mode.

However, the 354/388 nm UVAI is 0.5 higher than the 340/380 nm counterpart.

Satellite UVAI products have been widely used for dust plume tracking and source mapping, thanks to its ease of computation and great coverage over desert regions. A threshold UVAI value is usually used to isolate the dust signal. For example, Prospero et al. (2002) used a threshold of 1 for North Africa and 0.7 elsewhere. Schepanski et al. (2012) used a threshold of 2 to detect major dust plumes over the Sahara. Based on a UVAI threshold of 2, we find that EPIC has over 18% pixels exceeding the

threshold, 2 to 9 times more than other products (Fig. 3g). In contrast, using a percentile threshold better represents the dynamic range of each product and produces a more coherent detection of the dust plume. Specifically, using the 95th percentile as a threshold for dust detection, the products show a general agreement in delineating the areal extent of heavy dust plume from Aralkum. The plume features two major clusters of elevated UVAI values, separated by a cloud band (as seen in Fig. 3a). Another small cluster of large UVAI values was observed over the Tejen and Murghab river deltas in southern Turkmenistan,

likely due to local emission or transported dust from Afghanistan.

A persistent feature in all UVAI products is the presence of large positive values along the eastern coast of Caspian Sea, especially over the northern Caspian Sea and Garabogazköl gulf. To verify if airborne dust was responsible, we compare the CALIOP AOD with coincident UAVI products along the overpass shown in Fig. 3. We create the co-located data by first identifying the nearest pixels from all UVAI products to the CALIOP footprints, and then shifting the UVAI pixels along

track based on the sensor scan time differences. As shown in Fig. 3, CALIOP has an average time difference of 10, 6, and 23 minutes from OMPS, TROPOMI, and EPIC, respectively. The coincident AOD and UVAI retrievals are displayed in Fig. 4, after applying a moving average every 5 seconds to smooth the data.

CALIOP detected a dust layer at the southeastern coast of Caspian Sea (see Fig. 2d) with an AOD up to 1.8. Over the northern Caspian Sea and Garabogazköl gulf, CALIOP detected a very thin aerosol layer with negligible AOD, which may

be transported dust from the adjacent deserts (e.g., Ryn Desert). This thin aerosol layer, however, is unlikely to explain the large positive UVAI (i.e. >2) found in these regions. Such high UVAI values are found to persist during dust-free days and not change with time based on the high-cadence EPIC observations. Both the northern Caspian Sea and Garabogazköl gulf have an average water depth of less than 10 m, and contain large bodies of turbid or salty water (Modabberi et al., 2019). The



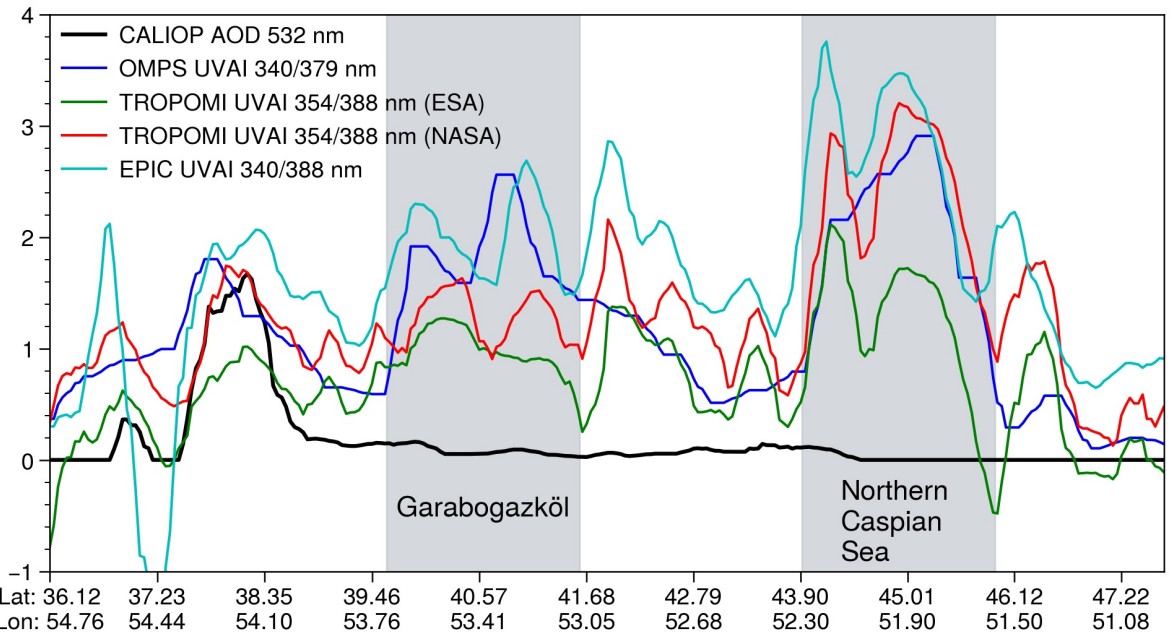

**Figure 4.** Comparison between CALIOP 532 nm AOD and UVAI products from OMPS, TROPOMI, and EPIC along the eastern coast of Caspian Sea on 27 May 2018. The Garabogazköl gulf and northern Caspian Sea are grayed out. The CALIPSO ground track is shown in Fig. 3.

persistent discoloring and turbidity of northern Caspian is caused by the influx of nutrients and sediments through the Volga
and Ural rivers (Moradi, 2022). The optically active constituents in the sediment-rich water are highly absorbing in the near UV, resulting in a distinct spectral signature (i.e. stronger absorption at shorter wavelengths) similar to that of airborne dust (He et al., 2012; Lee et al., 2013; Di Biagio et al., 2019). Similarly, the highly salty water of Garabogazköl gulf (i.e. with a salinity of about 35%) causes strong UV absorption and a dust-like UVAI signal, while the main body of Caspian Sea has a much lower salinity (1.3%) and produces near zero UVAI. In addition, the Sor Kaydak salt marsh to the east of northern Caspian features
large positive UVAI (Fig. 3), probably due to enhanced UV absorption by salt minerals and organisms (e.g., algae). Therefore, the enhanced UV absorption of turbid and saline waters along the Caspian coast causes the water-leaving UV radiances to deviate from that of a pure scattering atmosphere, resulting in large positive UVAI similar to the effect of airborne dust. This surface effect-related UVAI signal is expected to be common over ephemeral and dried saline lakes (including Aralkum Desert), presenting a possible source of errors in using UVAI products for dust detection over these potential dust source areas.

## 3.3 Comparison of AOD products

Due to the inhomogeneous and generally higher surface reflectivity, AOD retrieval over land involves larger inconsistencies in the algorithm treatment of aerosol microphysical properties and surface reflectivity compared to AOD retrieval over water.



**Table 2.** Summary statistics of eight AOD products in observing the Aralkum dust event of 27–29 May 2018, based on the best quality, over-land retrievals within region 47°E–70°E, 34°N–50°N. SD, standard deviation; MAD, median absolute deviation. P95, 95th percentile.

| AOD Product | N | Mean | SD | Median | MAD | Skewness | P95 | Max. |
|---|---|---|---|---|---|---|---|---|
| MODIS/Terra DB | 25,821 | 0.43 | 0.68 | 0.22 | 0.15 | 3.4 | 1.8 | 3.5 |
| MODIS/Aqua DB | 25,559 | 0.42 | 0.52 | 0.28 | 0.16 | 3.6 | 1.4 | 3.5 |
| MODIS/Terra MAIAC | 6,530,616 | 0.30 | 0.49 | 0.16 | 0.09 | 5.2 | 1.1 | 6 |
| MODIS/Aqua MAIAC | 6,209,626 | 0.33 | 0.63 | 0.15 | 0.08 | 4.7 | 1.3 | 6 |
| VIIRS/SNPP DB | 137,569 | 0.39 | 0.69 | 0.14 | 0.09 | 3.9 | 1.6 | 5 |
| VIIRS/NOAA20 DB | 133,199 | 0.49 | 0.80 | 0.20 | 0.14 | 3.5 | 2.0 | 5 |
| VIIRS/SNPP EPS | 6,179,831 | 0.44 | 0.59 | 0.21 | 0.16 | 2.5 | 1.8 | 5 |
| VIIRS/NOAA20 EPS | 5,311,729 | 0.35 | 0.52 | 0.16 | 0.14 | 2.8 | 1.5 | 5 |

The over-land AOD retrieval generally uses fewer spectral bands and more assumptions about region- and season-dependent aerosol conditions. In this section, we first evaluate the consistency of AOD products in observing the suspended dust over desert surfaces during 27–29 May 2018. Then, we compare the AOD products in observing the dust outflow to the Caspian Sea on 28 May 2018.

### 3.3.1 Dust over desert surfaces

We first report the statistical properties of eight AOD products, summarized in Table 2. The statistics are computed from the best quality, over-land retrievals from each product covering the Aral basin (47°E–70°E, 34°N–50°N) during the period of 27–29 May 2018. The number of pixels from each product varies greatly because of the different spatial resolutions, but also depends on the screening of irretrievable scenes (e.g., clouds, snow/ice, sun glint and ocean color) and the definition of retrieval quality or bias. These factors lead to inconsistent sampling by the products. The inconsistent sampling, together with different retrieval techniques (e.g., use of different aerosol optical models and surface reflectance database in the LUTs), contribute to the disparity in the AOD statistics. In addition to AOD algorithm differences, the NOAA EPS product uses L1b data from the NOAA VIIRS Calibration/Validation group, while the NASA DT and DB products use the L1b data provided by the NASA VIIRS Calibration Support Team.

Table 2 shows large variances (i.e. coefficient of variation>1) and strong positive skewness in all products, especially the MAIAC products. The MODIS regional mean (or median) AODs are more consistent between platforms than between algorithms. Specifically, the inter-platform (Terra vs. Aqua) difference in the mean AOD is 0.01–0.03 or less than 10%, while the inter-algorithm (DB vs. MAIAC) difference is 0.09–0.13 or more than 20%. For VIIRS, the inter-platform (SNPP vs. NOAA20) and inter-algorithm (DB vs. EPS) differences are comparable. The DB algorithm yields larger variances and skewness than EPS, indicating more extreme AOD values. We find that the EPS algorithm retrieves more frequent high AODs from VIIRS/SNPP than from VIIRS/NOAA20, causing significant differences in the regional mean AODs.





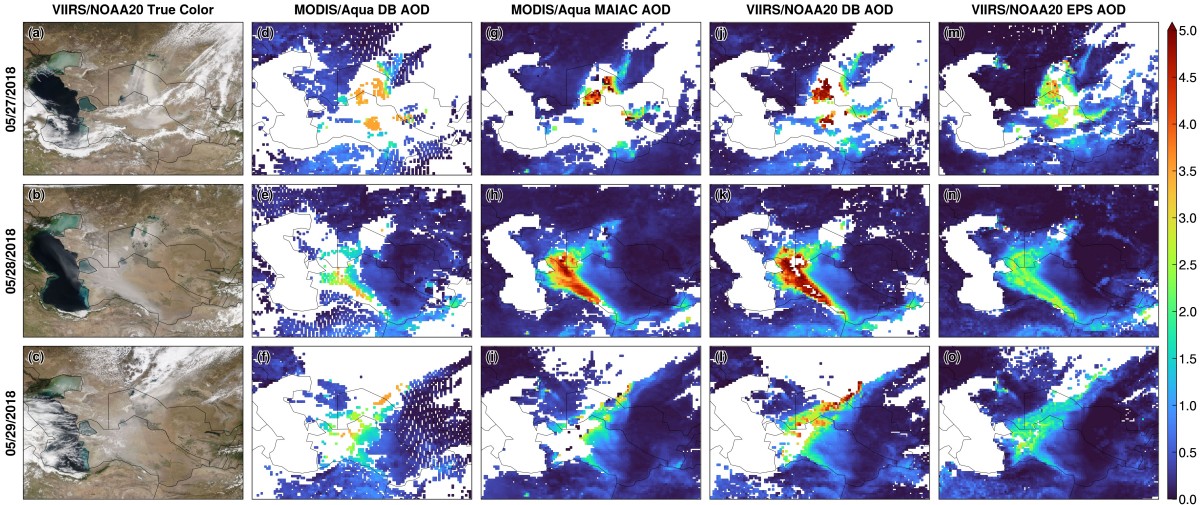

**Figure 5.** Multi-sensor AOD retrieval of the Aralkum dust storm on 27–29 May 2018: VIIRS/NOAA20 true color composite (a, b, c), MODIS/Aqua DB product (d, e, f), MODIS/Aqua MAIAC product (g, h, i), VIIRS/NOAA20 DB product (j, k, l), and VIIRS/NOAA20 EPS product (m, n, o). Only over-land retrievals are shown. All products are gridded to a 0.2°×0.2° resolution.

We further take a closer look at the DB and MAIAC products from MODIS/Aqua, and the DB and EPS products from
VIIRS/NOAA20. The instruments are hereafter referred to simply as MODIS and VIIRS unless otherwise specified. By comparing two different products from the same sensor, we can attribute the AOD differences to the choice of aerosol algorithms. As the AOD products are provided at different resolutions, we first regrid them onto a uniform 0.2°×0.2° grid, and calculate the mean value within each grid cell. The daily gridded AODs are displayed in Fig. 5, together with the VIIRS true color images which can help identify missing retrievals due to clouds or erroneous cloud screening.

All products failed to observe the full extent of the heavy dust plume on 27 May 2018. Among the four products, VIIRS EPS has the highest retrieval fraction. The fundamental approach of AOD retrieval from reflected sunlight relies on the brightening of the scene by aerosol scattering (primarily the forward scattering of surface reflection) against a relatively dark background. As the aerosol burden increases beyond a certain magnitude, the TOA reflectances start to flatten out and become insensitive to any additional increase in AOD. As a result, all algorithms prescribe an upper AOD limit, as shown in Table 2. Setting an
upper limit is necessary in order to reduce the risk of cloud contamination. The quality of retrieved AODs for heavy aerosol events tends to have large residuals and high possibility of cloud contamination. Indeed, we find that the missing retrievals in the VIIRS DB product were marked as marginal quality, and thus not included in the "best quality" estimate.

To verify if the freshly emitted dust plume on 27 May 2018 was erroneously categorized as clouds, Fig. 6 shows the cloud fraction or mask reported in selected L2 granules from each product. The TROPOMI TropOMAER UVAI is also shown to
indicate the dust plume location. Given the slight difference in overpass time between S5P and Aqua (5 minute apart) and between S5P and NOAA20 (1 hour apart), TROPOMI UVAI provides a nearly coincident view of the dust plume as MODIS



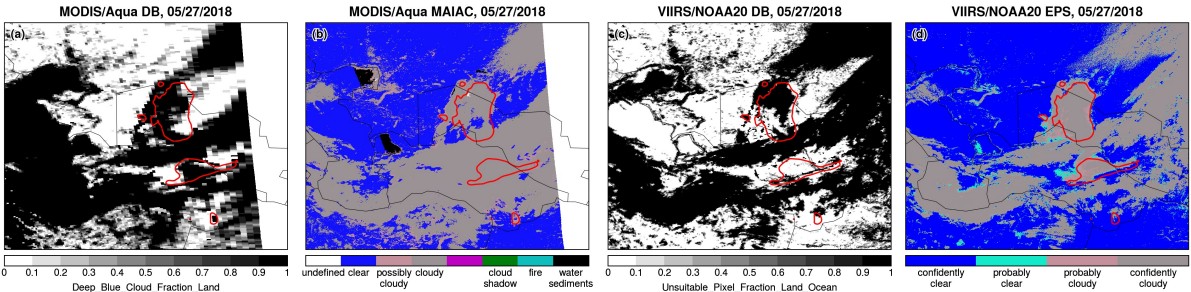

**Figure 6.** Mis-classification of the heavy dust plume as clouds on 27 May 2018. (a) Cloud fraction in the MODIS/Aqua DB product; (b) Cloud mask in the MODIS/Aqua MAIC product; (c) Unsuitable pixel fraction in the VIIRS/NOAA20 DB product; and (d) Cloud mask in the VIIRS/NOAA20 EPS product. The dust plume is indicated by the TROPOMI TropOMAER UVAI value of 2 (red contours).

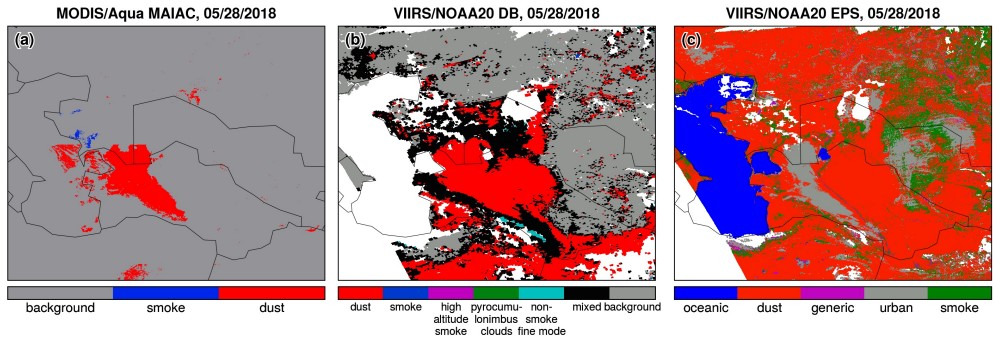

**Figure 7.** The aerosol optical model used for AOD retrieval on 28 May 2018 in the (a) MODIS/Aqua MAIAC, (b) VIIRS/NOAA20 DB, and (c) VIIRS/NOAA20 EPS products.

and VIIRS. Fig. 6 reveals that the majority of the heavy dust plume is classified as clouds in all products. For MODIS, the DB algorithm only detected the dust plume fringe area, where the retrieved AOD reached the algorithm limit (3.5). MAIAC classified the majority of dust pixels as "cloudy", but it was able to see the "clear" pixels through cloud gaps. Surprisingly,

MAIAC successfully detected the sediment-rich water of the northern Caspian Sea and Garabogazköl gulf. For VIIRS, the DB algorithm does not report the cloud fraction or mask, but instead report the fraction of level-1 pixels not used in the retrieval, named Unsuitable_Pixel_Fraction_Land_Ocean. This parameter shows that the dust pixels close to Aralkum were mostly excluded in the retrieval. Similarly, EPS classified most of the dust plume as "confidently cloudy", except for the plume fringe areas where the pixels are classified as "probably clear".

On 28 May 2018, the Aral Sea basin is influenced by a high pressure system that provided an ideal cloud-free condition for aerosol retrieval. Fig. 5 shows that all products were able to observe the extensive dust layer covering western Uzbekistan and Turkmenistan, albeit with large differences in the AOD magnitude. A striking difference among the products is that the VIIRS EPS product yields much lower (by more than 50%) AOD than others over the dust-affected region. As described





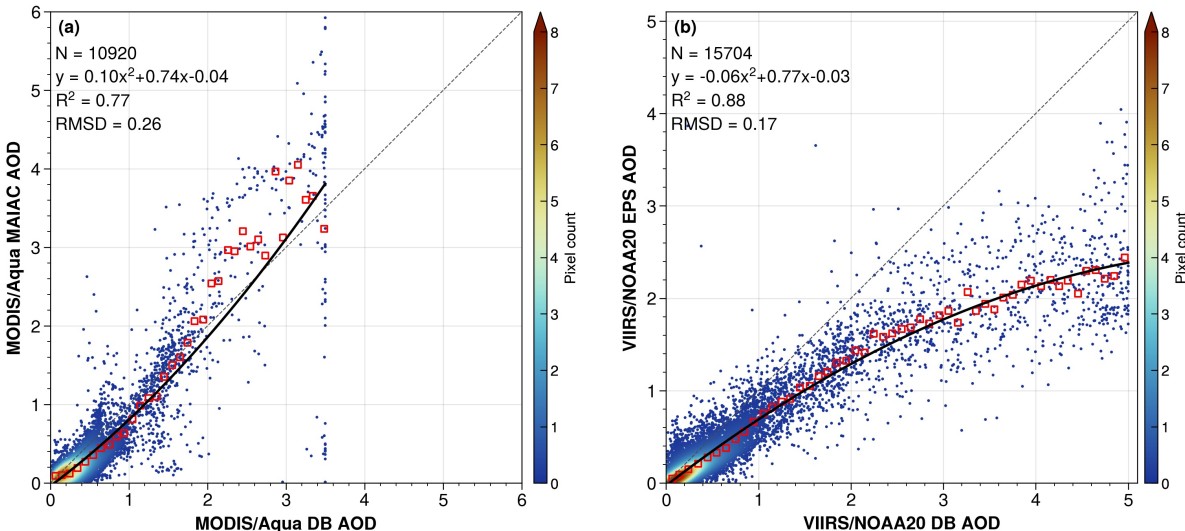

**Figure 8.** Comparison of over-land AOD retrieval using different algorithms during 27–29 May 2018: (a) MODIS/Aqua MAIAC vs. DB; (b) VIIRS/NOAA20 EPS vs. DB. Black lines indicate the quadratic model fits. Red squares indicate the bin-averaged AODs every 0.1 increments. RMSD, root mean square difference.

in Section 2, these aerosol algorithms use location- and season-dependent aerosol optical models for AOD retrieval over land, including a coarse-dominated dust model dedicated for dust retrieval. To find out whether the dust optical model was successfully used for retrieval on 28 May 2018, Fig. 7 displays the aerosol model used for retrieval within each product. The MODIS DB product does not report the aerosol model information and thus is not shown. The MODIS MAIAC and VIIRS DB products successfully selected their dust optical models for retrieval over the dusty scene in Turkmenistan. However, the VIIRS EPS product mistakenly selected the urban aerosol model for the dusty scene. According to Laszlo and Liu (2022), the EPS algorithm retrieves the AOD in the 0.41 $\mu$m channel first, which is then used to derive the AOD in other wavelengths based on the spectral normalized extinction coefficients determined by the selected aerosol model. In the EPS algorithm, the urban aerosol model yields a much stronger wavelength dependence in the extinction coefficient compared to the dust model (refer to Figure 3-4 in Laszlo and Liu (2022)). Assuming the same AOD at 0.41 $\mu$m, the dust model yields more than 40% higher AOD at 0.55 $\mu$m than the urban aerosol model. Therefore, had the dust optical model be forcibly used in the retrieval, the EPS AOD at 0.55 $\mu$m would have been significantly higher and brought into closer agreement with the DB and MAIAC algorithms.

Using the regridded AOD in Fig. 5, we further use regression analysis to examine the inter-algorithm AOD differences from MODIS and VIIRS. We find that the choice of algorithms for both MODIS or VIIRS causes a non-linear relationship between the AOD products. Hence a quadric model is used for fitting, as shown in Fig. 8. The model goodness of fit is measured by $R^2$ and root mean square difference (RMSD). Compared to DB, MAIAC produces lower AOD from MODIS under low aerosol burden, but higher AOD under heavy burdens. For example, the MAIAC product is 36% lower than the DB product when AOD<1, but 22% higher when 2<AOD<3. This nonlinear relationship may be partly explained by the higher resolution in





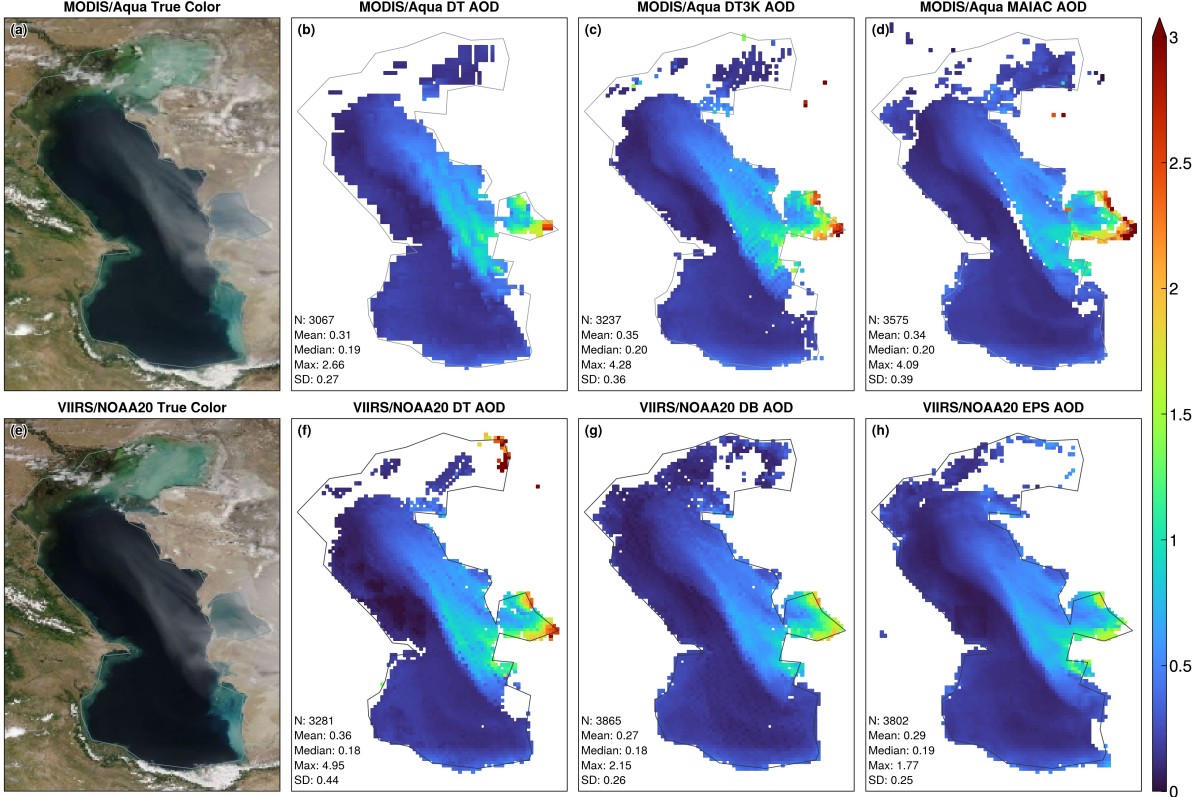

**Figure 9.** Multi-sensor AOD retrieval over the Caspian Sea on 28 May 2018. Top row panels are the MODIS/Aqua true color image (a), DT AOD (b), DT3K AOD (c), and MAIAC AOD (d). Bottom row panels are the VIIRS/NOAA20 true color image (e), DT AOD (f), DB AOD (g), and EPS AOD (h). All products are gridded to a 0.1°×0.1° resolution. The AOD summary statistics are shown on each panel.

the MAIAC product, which yields a greater number of pristine and heavily polluted pixels. The higher AOD limit in MAIAC (6 vs. 3.5 for DB) perhaps also contributes to a larger dynamic range and deviation from the DB product. For VIIRS, EPS generally yields lower AOD than DB under all conditions, with their difference increasing with the aerosol burden (Fig. 8b). For example, the EPS-DB difference increases from 37% when AOD<1 to 51% when 4<AOD<5. Indeed, Fig. 5 shows that EPS produces significantly lower AODs in dust-affected areas than other algorithms.

### 3.3.2 Dust outflow to Caspian Sea

Caspian Sea is affected by dust outflow from the adjacent deserts in Central Asia and distant sources in North Africa and the Middle East (Mohammadpour et al., 2022; Shukurov et al., 2023). As a source of micro-nutrients the dust outflow may contribute to intense algal blooms in the southern and southeastern parts of Caspian Sea where the marine productivity is limited by low nutrient levels (Modabberi et al., 2019; Kalinskaya et al., 2021). Fig. 9 displays the AOD retrieval over Caspian Sea on 28 May 2018, based on the best quality, over-water retrievals from MODIS (DT, DT3K, and MAIAC) and VIIRS (DT,



DB, and EPS). All six AOD products are regridded to a 0.1°×0.1° resolution. Bilinear interpolation is applied to the MODIS
DT AOD to fill the data gaps due to reduced pixel resolutions at the swath edges. All products observed the elevated aerosol

burden associated with dust outflow to the Garabogazköl gulf and eastern Caspian Sea. The three MODIS products show
generally good agreement in the regional statistics. The VIIRS DT product show extremely large AOD over northern Caspian
Sea, where the turbid water likely violates the "dark target" assumption in the over-water retrieval. The non-negligible water-
leaving radiance from these pixels may be erroneously treated as aerosol signal, resulting in AOD overestimation compared to
the VIIRS DB and EPS products.

The products are further compared via linear regression to examine the consistency between aerosol algorithms, as shown
in Fig. 10. Compared to the over-land retrieval, the over-water retrievals show a better agreement in the AOD magnitude and
strong linear correlations, with the $R^2$ above 0.9 and RMSD below 0.1 for all comparisons. In particular, the MODIS DT and
DT3K products yield a slope of 0.98 and $R^2$ of 0.91, indicating excellent agreement. The linear model fits suggest that the
DT algorithm yields smaller AOD under clean marine conditions (i.e. AOD<0.15), and higher AOD over dust-affected areas

compared to the MAIAC, DB, or EPS algorithms.

   All over-water AOD algorithms assume a linear combination of one fine and one coarse mode in representing the ambient
aerosol, and simultaneously retrieves the total AOD and FMF. Here we use the total AOD and FMF to compute the coarse-mode
AOD from each product, which is then regridded to a 0.1°×0.1° resolution. The regridded coarse-mode AODs are compared via
linear regression, as shown in Fig. 11. The coarse-mode AODs are strongly linearly correlated, but show larger inter-algorithm

differences than the total AODs. In general, the DT algorithm tends to produce higher coarse-mode AOD under dust-laden
conditions, compared to other algorithms.

## 3.4 Comparison of AOCH products

Figure 12 displays several parameters from the EPIC AOCH product on 28 and 29 May 2018, including the 680 nm surface
reflectivity, 680 nm AOD, AOCH, and AOCH after applying a filter of AOD>1.8. The heavy dust plume on 27 May 2018 was

missed by EPIC and thus not shown. On 28 May 2018, the low surface reflectivity of Caspian Sea (i.e. below 0.03), combined
with the elevated AOD (i.e. 0.5–1) associated with the dust outflow, provides a suitable condition for inferring AOCH from the
DOAS ratios in the oxygen A and B bands. Fig. 12e shows that the dust layer was elevated to an altitude of 1.5–2.5 km over
the Caspian Sea, probably due to the orographic lifting by the Ustyurt Plateau.

   Unlike the Caspian Sea, the surface reflectivity exceeds 0.18 over the desert surfaces surrounding the Aral Sea. Through

radiative transfer simulations Xu et al. (2019) showed that the sensitivity of DOAS ratios to AOCH started to diminish over
bright surfaces, and that a high aerosol burden is needed to produce sufficiently strong aerosol signals for AOCH to be retriev-
able from the DOAS measurements. Fig. 12 shows that on both days EPIC retrieved extremely large AODs (up to 6 at 680 nm)
associated with the lofted dust over Turkmenistan and low AODs elsewhere. The retrieved AOCH shows an opposite pattern,
with low values over high aerosol burden areas and unrealistically high values over low aerosol burden areas. Therefore, we

will employ the 680 nm AOD to filter the unreliable AOCH retrieval associated with weak aerosol signals over desert areas.





**Figure 10.** Comparison of AOD products using different algorithms over Caspian Sea on 28 May 2018: (a) MODIS/Aqua DT3K vs. DT; (b) MODIS/Aqua MAIAC vs. DT, (c) VIIRS/NOAA20 DB vs. DT, (d) VIIRS/NOAA20 EPS vs. DT. Black lines indicate the linear regression fits. Red squares indicate the bin-averaged AODs every 0.05 increments. RMSD, root mean square difference.

By applying a filter of AOD>1.8, we find that the majority of the unrealistically high AOCH pixels are removed, resulting in a spatially continuous AOCH pattern from Turkmenistan to the Caspian Sea (Fig. 12g). The AOD-filtered AOCH suggests that the lofted dust over Turkmenistan was located between 1–1.5 km on 28 May 2018, and between 2–3 km on 29 May 2018, or twice as high as the previous day. The temporal variation of dust height can be explained by the meteorological condition:

a prevailing high pressure system may have suppressed the vertical aerosol mixing on 28 May 2018, whereas the enhanced upward vertical motion ahead of a deepening cold front promoted the convective mixing of the lofted dust on 29 May 2018.







**Figure 11.** Same as Fig. 10 but for the coarse-mode AOD.

Using the daytime CALIOP overpass on 29 May 2018, we evaluated the accuracy of EPIC AOCH against the CALIOP aerosol extinction-weight height, and explored the effects of wavelength choice (532 vs 1064 nm) and AOD filter. The CALIPSO ground track is shown in Fig. 12f. Past studies suggested that the CALIOP 1064 nm measurement is more sensitive to coarse aerosols close to the ground, while the 532 nm channel is subject to sensitivity loss in the presence of carbonaceous aerosols (Torres et al., 2013). Fig. 13a shows that the choice of 532 or 1064 nm has a minor effect on the CALIOP extinction-weighted height, with a mean difference below 0.1 km or 5%. Both indicate that the dust layer was mostly located between 1–2 km.



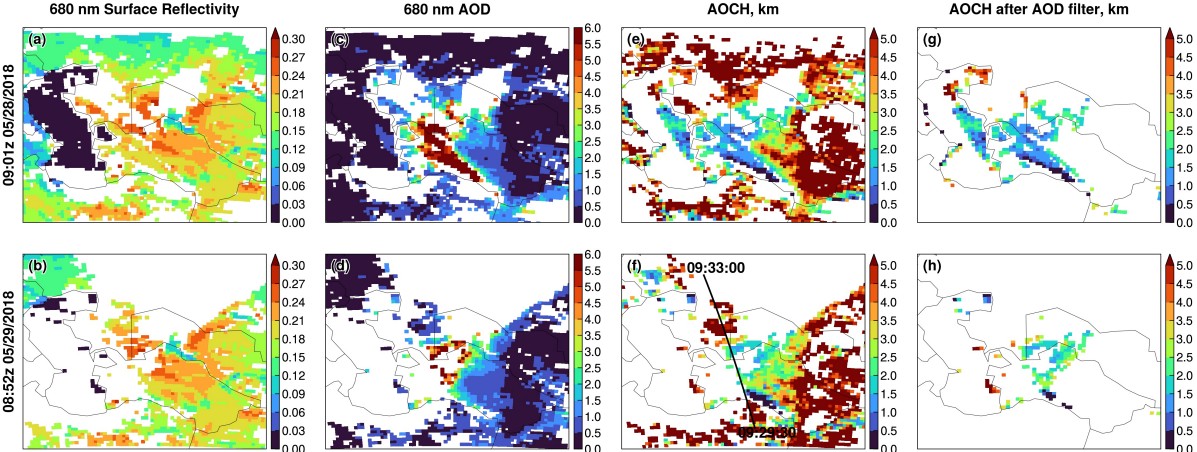

**Figure 12.** The 680 nm surface reflectivity (a, b), 680 nm AOD (c, d), AOCH (e, f), and AOCH after applying a filter of AOD>1.8 (g, h) based on the EPIC AOCH product at 09:01z 28 May 2018 (top row) and 08:47z 29 May 2018 (bottom row). The daytime CALIPSO ground track on 29 May 2018 is shown in panel (f).

The EPIC AOCH has good agreement with CALIOP over the dusty or high AOD areas, but shows significant overestimation
elsewhere. Using the CALIOP 532 nm extinction-weighted height as ground truth, Fig. 13b shows the mean bias and root-mean-square-error (RMSE) of the EPIC AOCH for a range of threshold AOD filters. EPIC persistently overestimates the aerosol height, especially under low aerosol burdens. The mean bias and RMSE are greatly improved and both fall below 0.5 km, when an AOD threshold of 1.8 or above is used to filter the EPIC AOCH. In other words, the EPIC AOCH product has the best agreement with CALIOP over regions affected by heavy dust burden (i.e. AOD>1.8), where the mean bias and RMSE
are comparable to the over-water AOCH retrieval as reported in previous studies (Xu et al., 2017, 2019). When using the 1064 measurement as ground truth, we find that the mean bias is slightly improved (0.25 vs. 0.3 km), while the RMSE is slightly degraded (0.58 vs. 0.42 km).

## 4 Conclusions

Driven by the desiccation of the Aral Sea, the newly formed Aralkum Desert has emerged as a major source of wind-blown
saline dust aerosol in Central Asia. There is generally a poor understanding of the performance and consistency of satellite aerosol remote sensing techniques in observing the airborne dust from Aralkum Desert. This paper addresses this knowledge gap in two aspects.

First, we provided a review of the physical principles and algorithms used to retrieve UVAI, AOD, and AOCH over desert surfaces, focusing on the algorithms' treatment of regional aerosol optical properties and land surface reflectivity. As satellite
remote sensing aerosol algorithms are generally optimized for global performance, the algorithm differences are expected to have a larger effect on the product performance and consistency over Central Asia, where the local aerosol and land surface



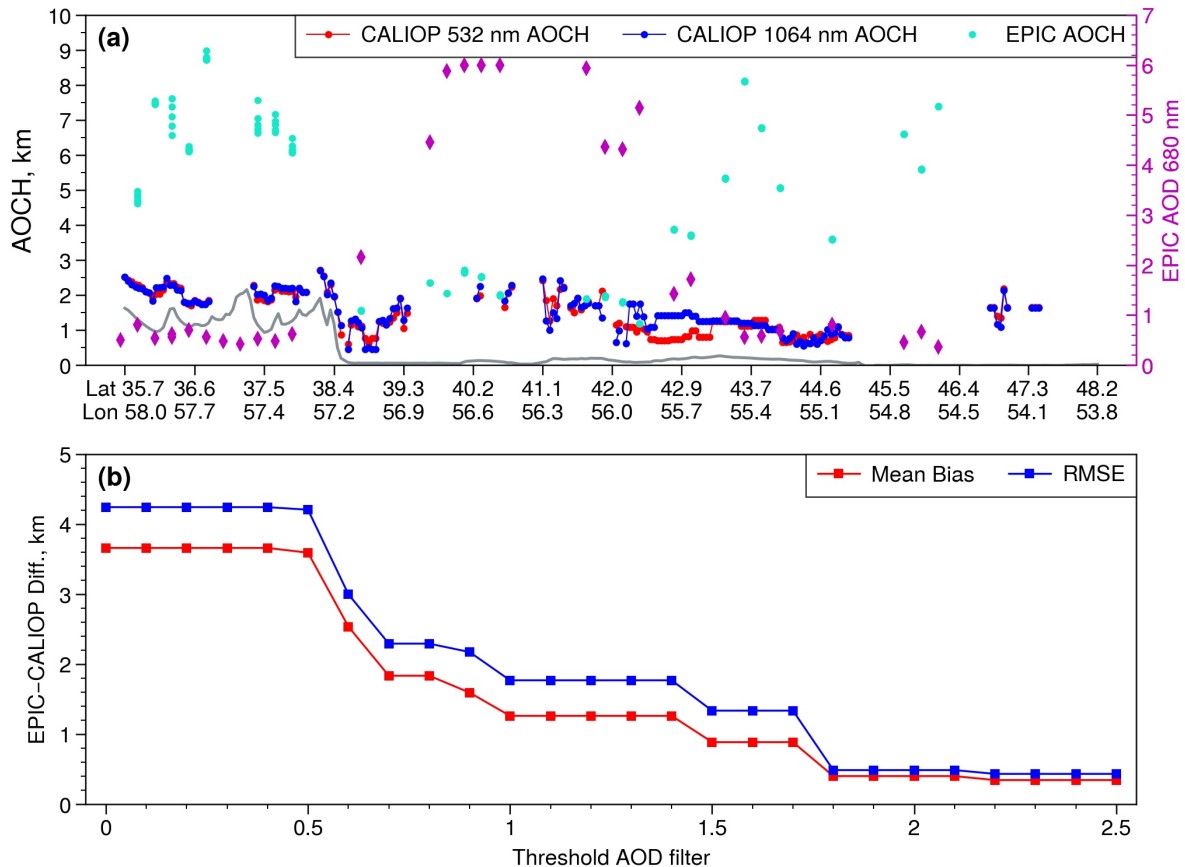

**Figure 13.** (a) Comparison between CALIOP 532 and 1064 nm aerosol extinction-weighted heights (dotted lines) and EPIC-retrieved AOCH (cyan dots) along the CALIOP overpass on 29 May 2018 (ground track shown in Fig.12f). The EPIC-retrieved 680 nm AOD (magenta diamonds) and CALIOP-detected ground surface (gray line) are also shown. (b) The mean bias and RMSE of EPIC AOCH against CALIOP 532 nm extinction-weighted height as a function of the threshold AOD used to filter the EPIC AOCH retrieval.

conditions are under-represented in the aerosol algorithms. For example, the dust optical models used in the AOD and AOCH algorithms are mostly derived from AERONET measurements over West Africa and the Middle East. The extent to which the dust optical models deviate from the saline dust from Aralkum (e.g., refractive index, size distribution) has yet to be determined. 570 The impact of the different algorithm assumptions on aerosol product performance in Central Asia is unknown.

Second, we presented a case analysis of the cross-sensor and cross-algorithm consistency in observing a saline dust storm from the Aralkum Desert during 27–29 May 2018. We considered a wide range of aerosol products, including the UVAI products from OMPS, TROPOMI, and EPIC, the AOD products from MODIS and VIIRS based on multiple algorithms (DT, DT3K, DB, MAIAC, and EPS), and the AOCH derived from CALIOP aerosol extinction profile and EPIC oxygen absorption 575 spectroscopic measurements. The main findings are as follows.



The UVAI products show similar spatial patterns associated with the heavy dust plume on 27 May 2018, but reveal significant differences in magnitude and dynamic range. The differences can be explained, at least in part, by the choice of wavelength pair and the treatment of cloud scattering. Using the 95th percentile of each product as the threshold for dust detection, we find a general agreement between the products in delineating the areal extent of the dust plume. All UVAI products show large positive values over the northern Caspian Sea, Garabogazköl gulf, and Sor Kaydak salt marsh. This dust-like signal is primarily caused by enhanced UV absorption by turbid and saline waters, which causes the water-leaving UV radiances to deviate from a pure Rayleigh scattering atmosphere, similar to the effect of absorbing aerosols. Hence, caution must be used to avoid misinterpreting the surface effect as dust signal over ephemeral or dried water bodies, such as the Aralkum Desert.

The AOD products over desert surfaces are subject to significant inconsistency between different sensors and algorithms. The choice of aerosol algorithms causes a nonlinear response in the retrieved AOD from MODIS and VIIRS. Specifically, the MAIAC algorithm yields lower AOD than the DB algorithm under low aerosol burden, but higher AOD under heavy aerosol burden. The EPS algorithm yields lower AOD than DB under all aerosol burden conditions. The EPS-DB difference is found to increase with the aerosol burden. In addition, all AOD products erroneously classified the freshly emitted dust plume on 27 May 2018 as clouds. The EPS algorithm misused the urban aerosol optical model for dust retrieval, which may explain the significantly lower AOD in the VIIRS EPS product compared to other products.

There is generally good agreement between the over-water AOD retrievals in observing the dust outflow to the Caspian Sea. The AOD products show strong linear correlations and low RMSD, with the $R^2$ above 0.9 and RMSD below 0.1. We find that the DT algorithm tends to yield smaller AOD under clean marine conditions, but higher total and coarse-mode AOD under dust-laden conditions compared to the MAIAC, DB, and EPS algorithms.

Despite the high surface reflectivity which limits the passive AOCH retrieval over Central Asia, we find that the EPIC AOCH product retrieved the dust layer height reasonably, after applying an AOD filter to remove the unrealistic retrievals under low aerosol burdens. Specifically, the EPIC AOCH product shows the best agreement with CALIOP aerosol extinction-weighted height over areas with AOD>1.8, where both the mean bias and RMSE are below 0.5 km. However, the EPIC AOCH product significantly overestimates the aerosol height over regions with low AOD burdens. The transported dust from Aralkum to Turkmenistan was located at an altitude of 1-1.5 km on 28 May 2018 due to a prevailing high pressure system, and lifted to an altitude of 2-3 km the next day, due to enhanced vertical mixing ahead of an advancing cold front. The dust outflow to the Caspian Sea was lifted to an altitude of 1.5–2.5 km, likely due to the orographic effect of the Ustyurt Plateau.

In summary, the dust event analysis allows us to conduct a detailed comparison of various satellite products in characterizing the airborne dust from the Aralkum Desert. The analysis reveals a potential limitation of using UVAI for dust detection and source mapping over Central Asia, due to the interference effect of turbid/salty waters, salt marshes, and saline deserts. While this study reveals substantial inconsistency between the AOD products, an extensive analysis (e.g., using multiple years' data) is needed to provide a more robust evaluation of the product differences in the retrieval fraction, climatological properties, and long-term trend. This study highlights the need for in situ measurements of the physiochemical and mineralogical properties of the saline dust from Aralkum, which are critical for improving the representation of the regional aerosol optical models used in satellite aerosol algorithms. We also call for routine ground-based aerosol measurements in the downwind region



(e.g., Karakalpakstan) to provide ground truth data for validating the satellite aerosol products. Finally, while the case study demonstrates the use of EPIC AOCH product to quantify the dust layer height with a reasonable accuracy, an extensive analysis is desirable to evaluate the information content and skill of passive techniques in retrieving the aerosol vertical height over desert areas, particularly the role of surface reflectivity and aerosol optical properties.

*Data availability.* All the data used in the study are publicly available. The data repository links are provided in the reference column of Table 1. The regrided daily AOD products are available at https://doi.org/10.5281/zenodo.13994593.

*Author contributions.* XX designed the study and performed data curation, formal analysis, visualization, and writing of the initial draft. JW contributed to the writing of EPIC AOCH algorithm. IL contributed to the writing of VIIRS EPS algorithm. CA and OT contributed to the UVAI analysis. AS, JL, RL, YW, AL, and IL contributed to AOD analysis. JW, ZL and OT contributed to the AOCH analysis. All authors
edited the manuscript.

*Competing interests.* The authors declare no competing interests.

*Acknowledgements.* XX acknowledges support from the NASA Land-Cover and Land-Use Change (LCLUC) Program (grant no. 80NSSC20K1480). The satellite data teams acknowledge funding support from multiple NASA and NOAA funding sources supporting the multi-decadal development and evaluation of these data records. The authors gratefully acknowledge the Copernicus Data Space Ecosystem and NASA ASDC,
LAADS DAAC, and GES DISC for maintaining/distributing the data products used in this study.





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
