# Peer review of "Analysis of a saline dust storm from the Aralkum Desert - Part 1: Consistency between multisensor satellite aerosol products"

_EGUsphere, 2024_

## Author Comment (AC1)

On behalf of all authors, I would like to thank the reviewers and Dr. Sophie Vandenbussche for their careful reading and comments of this manuscript. In addition to minor editing, we have added the following new materials:

1. Better explanation of this study's objective. This paper focuses on evaluating the consistency, rather than the performance or accuracy, of **operational satellite products** in observing the saline dust from Aralkum. Due to limited time and resources (and lack of in situ measurements), we are unable to perform extensive algorithm refinements for this case study, which by itself is non-trivial work and worthy of a dedicated effort. In response to reviewers' comments, however, we added new results on VIIRS EPS AOD using a dust optical model and the MISR standard and research algorithm retrievals.

2. Used the updated TROPOMI UVAI product from ESA which includes the 335/367 nm pair.

3. Added descriptions of MISR aerosol algorithms (Section 2.2.5, 2.4.4), and analysis of MISR results from both the operational and research algorithms (Section 3.3.1).

4. Added descriptions of thermal-infrared aerosol algorithms (Section 2.3, 2.4.3) and analysis of four IASI AOD products using the LMD, ULB, MAPIR and IMARS algorithms (Section 3.4).

5. Added results on IASI mean dust layer altitude and MISR plume height (Section 3.5).

Detailed responses to reviewer comments are provided below.

Reviewer #1:

Review of Analysis of a saline dust storm from the Aralkum Desert - Part 1: Consistency of multisensor satellite aerosol products by Xin Xi

This paper describes a dust outbreak over the Aralkum desert, claiming to address a knowledge gap in the representation of satellite observations of saline dust emissions. Several established aerosol products, like UVAI and AOD from NASA and NOAA sensors are compared, along with somewhat newer aerosol products like MAIAC AOD and aerosol optical centroid height (AOCH) from EPIC, compared with AOCH from CALIOP.

The first part is extensive and reads like a review of current satellite remote sensing techniques, but focuses only on NASA/NOAA single view instruments with their known limitations, which are confirmed in this study. In section 2 a review is given, suggesting a comprehensive description of Aerosol Remote Sensing of Desert Areas. However, the description is general in the sense that much is touched upon, but also leaves out many details for a full review, so that it is unclear if it is meant to be specific for this case, or a complete description for the interested reader. E.g. the UVAI description repeats information from the late 90s and describes no new developments. The different algorithms, cloud-corrected or not, is not described. It claims that 'historically, two wavelength pairs have been used", but does not mention the reason of channel availability for the 340/380 nm choice (and some other wavelength pairs not mentioned) and the important

more optimal choice of 354/388 nm outside ozone absorption bands that can be implemented for spectrometers. Furthermore, OMI is described briefly, but not used in this study. On the other hand, different TROPOMI UVAI products are described and compared, but not which versions. The authors claim that two wavelength pairs are available. Since version 2 of TROPOMI L2 products, the ESA TROPOMI UVAI products also has a 335/367 nm pair available, outside ozone absorption, in preparation of the Sentinel-5 mission. Only in the version 1 data two UVAI wavelength pairs were used, which raises the question what version the authors have used and whether they are consistent. Version 1 was based on collection 1 L1b data, while version 2 is based on L1b data that was degradation-corrected. It is unclear whether the NASA and ESA TROPOMI UVAI products that are compared are even based on the same L1b input.

AR: Section 2 is not meant to be a comprehensive review, but focuses on the techniques capable of retrieving aerosol properties over Central Asia. In the review, we focus on each algorithm's theoretical basis, limitations, and assumptions of aerosol and surface properties. We are hoping that the review will give interested readers (e.g.,researchers in Central Asia) a primer on aerosol remote sensing, particularly the different techniques and a priori assumptions which may deviate from actual conditions in Central Asia.

Thanks for pointing out the 335/367 nm wavelength pair in the ESA TROPOMI UVAI product. We have updated our analysis with the version 2 reprocessed product which includes the 335/367 nm pair.

The AOD description is also elaborate, citing quite extensively from papers of the last two to three decades with seemingly random details. If anything, the section gives a good impression of the despair and indecision a naive user feels when seeing the sprawl of AOD products. Instead of making clear choices from the different AOD products that NASA and NOAA have developed, the products are put together in files, leaving it up to the user to decide what is best to use. The co-authors of this paper are the algorithm developers of the products and the absolute world-leading experts in their field, but even after this study the only conclusion is that AOD products are mainly inconsistent.

AR: Central Asia is under-sampled by ground aerosol monitoring networks like AERONET. This, combined with the complex underlying surfaces, are expected to cause larger disagreement and biases in satellite products in Central Asia compared to other better-observed regions (e.g., North Africa and Middle East). We believe this study is the first to compare such a broad range of products for Central Asia. Unfortunately, because of the lack of validation data, we are unable to rank the product performance. However, the synergistic multisensor analysis does reveal some limitations and biases in the products.

Having said that, the analysis of the performance of the satellite products over the Aralkum desert is decent. One of the interesting findings is the use of the 95 percentile of the UVAI to indicate a plume. The percentile is much less dependent on the wavelength choice or calibration or UVAI definition than a threshold method, which is an interesting find. Unfortunately, the authors do not give any recent references to the use of the UVAI as an indicator of plumes, even though the UVAI has been quite successful for that (e.g. Khaykin et al, 2022, Nature comm.).

The ever recurring discussion is what threshold should be used, while using a percentile could be more consistent, which is unfortunately not further explored. Even more disappointing is the fact that the authors revert to a threshold method in their own analysis!

AR: The choice of UVAI threshold for dust detection is arbitrary. A single case study is NOT sufficient to identify or recommend a universal threshold that can be applied to all sensors or regions. In fact, our analysis shows that one cannot apply a universal, fixed threshold value for dust detection, but should consider the dynamic range of each product. Our results show the percentile-based threshold produces more coherent dust detection between different UVAI products, whereas using a fixed threshold (e.g., 2) would have resulted in a large disagreement in the dust plume extent.

We did not revert to the fixed threshold in Fig. 6. The 95% percentile is actually equal to 2 for the NASA TROPOMI UVAI product. We have revised the wording to avoid misunderstanding.

My main criticism is that no effort has been done to try and improve the AOD retrievals with the knowledge of the dust emissions. It is known that AOD retrievals suffer from (inaccurate) knowledge of aerosol type. Here, an extensive knowledge of dust type is claimed, judging form Fig 1., but none of this information is used. One would expect at least some tests using a more proper aerosol model to quantify the improvement that can be expected, or a choice of AOD product that is most suitable. Instead, the authors conclude "HAD the dust optical model be forcibly used in the retrieval, the EPS AOD at 0.55 μm WOULD have been significantly higher and brought into closer agreement with the DB and MAIAC algorithms." which is too general a statement and not new, to say the least. Who else should perform such an analysis then the algorithm developers? With missions like PACE and EarthCARE it can be expected to have better aerosol microphysical properties available in the near future, and it would be interesting to know what can be expected for the AOD products described here, and what is needed to improve them.

AR: Please see our responses above. We mainly focus on global operational products, and the inconsistencies between them. In response to this comment, however, we added new results on NOAA EPS retrieval using a dust optical model, as well as MISR research algorithms with more realistic aerosol models than the MISR operational standard product.

I conclude that the paper may be interesting for readers with 'a knowledge gap in the performance and consistency of (NASA/NOAA) satellite observations in characterising the saline dust emission from the Aralkum Desert', but I am not impressed.

Specific comments:

line 58: . Data users may struggle with the product choice, not knowing the strengths and limitations of different products when applied to their region of interest.

Unfortunately, no recommendations are provided for users after this study. Nor are QA values updated or created as a result from this study. One result is that the AOCH is reliable for AOD> 1.8, which is a very high number. It is unlikely that users would benefit from such a threshold.

AR: Unfortunately, a single case study without ground truth is far from sufficient to draw conclusions on product performance or make recommendations on the QA rules. The EPIC AOCH is a new product, and one of only a few with the capability to retrieve aerosol heights over deserts. We have updated the CALIOP ALH results and EPIC validation. We use the 680 nm AOD threshold as a practical means to remove marginal quality retrievals. The choice of AOD threshold can vary by location and time. The thresholds we chose in this study are meant to minimize the effects of low signal-to-noise ratios, and should not be taken as a universal rule.

line 99: Historically, two wavelength pairs have been used: 340/380 nm and 354/388 nm.

Historically, more different wavelengths pairs have been defined, depending (mainly) on the instrument capabilities. With the introduction of hyperspectral instruments including the UV, the wavelengths became an actual choice, so the ozone-free wavelengths 354/388 nm were mostly chosen, next to the most used 340-380 nm. I suggest to add this to the review.

AR: Thanks for pointing this out. In the TROPOMI section, we added the following: "*The ESA product version 3 reports LER-based UVAI at three wavelength pairs: 354/388, 340/380, and 335/367 nm. The first two pairs were selected to continue the multidecadal heritage UVAI records from previous missions, while the 335/367 nm pair was added in 2022 to ensure compatibility with the future UVAI algorithm planned for Sentinel 5*."

line 146:

ESA TROPOMI UAI is in three wavelength pairs. Which version has been used? version 2 UVAI is about 0.5 higher than version 1 data for the displayed area, due to degradation correction. What version of data is used for the NASA UVAI?

AR: Thanks for pointing this out. We have updated the analysis using the latest TROPOMI AER_AI reprocessed version 2 product, which includes the new 335/367 nm pair. We find that "*The choice of wavelength pairs has a minor effect on the statistical distribution shape of the ESA TROPOMI products. The 354/388 nm UVAI is about 0.1 higher than the 340/380 nm pair. In comparison, the 335/367 nm pair produces less dispersion but significantly lower values, indicating reduced sensitivity to aerosol absorption*."

2.1.2 OMI UVAI is not used, why is it described? Either describe all available products, or the ones used in this study.

 AR: To avoid confusion, we have removed the OMI algorithm section.

158: AOD provides aerosol burden in the atmospheric column ->

"AOD is the aerosol light extinction of the total atmospheric column" A proper physical definition is not out of place in an atmospheric physics paper.

AR: Agreed. It has been revised to "*AOD is a dimensionless measure of light extinction due to aerosol scattering and absorption in an atmospheric column*".

line 212: strange sentence, please rephrase in a way that does not suggest a self-replicating algorithm.

AR: Satellite Ocean Aerosol Retrieval or SOAR is the official name for the algorithm. The algorithm was previously applied to SeaWiFS (Sayer et al. 2012), but has since been ported to MODIS and VIIRS (Sayer et al. 2018). We have revised the sentence to: "*DB has been expanded to all cloud-, snow- and ice-free land surfaces, and also performs retrieval over water using the Satellite Ocean Aerosol Retrieval (SOAR) algorithm*".

line 248: Once dust is detected -> If dust is detected

AR: Fixed.

line 324 Six EPIC bands are considered, including the oxygen A and B bands at 764 and 688 nm and two reference continuum bands at 780 and 680 nm. These are four (described) bands.What are the other two and why are they left out?

AR: EPIC used blue (443 nm) and red (680 nm) bands to retrieve AOD, $O_2$ A (764 nm) and B (688 nm) bands for retrieving AOCH and 2 UV bands (340 and 388 nm) for calculating UVAI. We have updated the text.

line 346: "they are subject to larger biases and usually not recommended for use in scientific studies. "

Wow! If scientific user shouldn't use them, who should? Non-scientific users?

AR: AE and SSA retrievals over land are more uncertain than the AOD retrieval, and because they are strongly dependent on the aerosol optical model selected during retrieval, they provide limited information about the aerosol size or type.

line 398 UAVI = UVAI

AR: Fixed.

line 398-400: "We create the co-located data by first identifying the nearest pixels from all UVAI products to the CALIOP footprints, and then shifting the UVAI pixels along track based on the sensor scan time differences.

I don't understand this. Please, elaborate.

AR: There are slight differences in the scan times (<30 min) between CALIPSO and UVAI sensors. To correct for the time differences, we shifted the UVAI products along-track to match the UVAI and AOD maxima observed at the southeastern coast of Caspian Sea.

line 455: The quality of retrieved AODs for heavy aerosol events tends to have large residuals and high possibility of cloud contamination.

I don't understand this sentence, please, rephrase.

AR: This section has been revised to "*None of the algorithms captured the full extent of the fresh dust plume on 27 May 2018, while the retrieved AOD for the dust scene reached the upper limit of each algorithm. There are several possible reasons. First, the AOD retrieval from reflected sunlight relies on the scene brightening by aerosol scattering (primarily the forward scattering of surface reflection) against a relatively dark background. At high aerosol loadings, the TOA reflectance becomes less sensitive to additional AOD increases. Second, AOD retrieval is performed by matching observed TOA reflectances against pre-computed values within a specific AOD range, which may be a poor fit if the aerosol burden is high or exceeds the predefined AOD limit. These retrievals are often flagged as marginal quality, and consequently excluded from our screening for best quality data. Setting an upper AOD limit is necessary to minimize the impact of residual cloud contamination. Heavy dust scenes may be either retrieved but with low confidence in clear sky, or classified as clouds and not retrieved.*"

Caption fig 6: MAIC = MAIAC

AR: Fixed.

Caption Fig 6: The dust plume is indicated by the TROPOMI TropOMAER UVAI value of 2 (red contours). Why not the 95th percentile? I thought this worked better and was more consistent.

AR: We changed it to 95th percentile. In fact, the 95th percentile is equal to 2.

line 507. If "the non-negligible water-leaving radiance from these pixels may be erroneously treated as aerosol signal" in the VIIRS DT algorithm, why is MODIS DT not affected?

AR: These pixels are not from the northern Caspian Sea, but the nearby salt flat. They were missed probably due to incorrect land/water masks. These are highly variable environments and may not be kept up-to-date in the geolocation map.

line 517: retrieves -> retrieve

AR: Fixed

line 550: Using the CALIOP 532 nm extinction-weighted height as ground truth"

Caliop is space-based.

AR: Changed to "validation data".

line 557: degraded -> Reduced

AR: Fixed.

lines 569-570: The extent to which the dust optical models deviate from the saline dust from Aralkum (e.g., refractive index, size distribution) has yet to be determined.

Why has this not been addressed in this paper?

AR: Because no data exists for such a challenging environment. And that is a main motivation for this study to test the satellite algorithms for such challenging aerosol and surface conditions.

Line 589-590: Why was the EPS algorithm not tested with the correct optical (dust) model? Would this improve the situation? Who else should test this? What is the use of a focus on a desert dust case with measurements on the dust properties and not using those measurments?

AR: We ran a test retrieval using the dust optical model. It improves the AOD magnitude, but also marks some retrievals as "marginal quality" because they reach the upper AOD limit and requires extrapolation in the spectral fitting, hence marked as low quality and excluded from the QA screening.

line 610: "This study highlights the need for in situ measurements of the physiochemical and mineralogical properties of the saline dust from Aralkum, which are critical for improving the representation of the regional aerosol optical models used in satellite aerosol algorithms.

We also call for routine ground-based aerosol measurements in the downwind region..."

The measurements available now are not used and the paper doesn't even show the possible improvement using a better dust representation. I suggest to start with that.

AR: As far as I know, there are no in situ data on the particle size distribution and refractive index about saline dust from Aralkum. Also, satellite aerosol algorithms are optimized to maximize global performance by capturing the most representative aerosol conditions. That means, the conditions in Central Asia are likely underrepresented in operational algorithms.

Reviewer #2:

This paper documents analysis of aerosol properties over the newly formed Aralkum desert using multi-sensor satellite observations. Such study is indeed important as aerosol properties from this desert contain significant amount of salt as well as dust, which is unique and maybe be well represented by satellite retrieval algorithms. However, I read through the manuscript but was disappointed because nowhere in the paper discussed the difference between aerosol properties here and other desert regions. To me, this is the most important point of such a study.

So I have only one major comment, which is, compare the optical properties of saline dust storms with regular dust storms. This can be done through:

AR: The objective of this paper is to investigate the inconsistency and potential biases of operational satellite aerosol products in characterizing the saline dust from Aralkum. As the reviewer pointed out, the saline dust is likely not well represented by the candidate aerosol optical models considered in satellite algorithms. But, in situ aerosol measurements (e.g., AERONET) near Aralkum are nonexistent, and thus no data are available to either constrain the assumptions in satellite algorithms or validate the retrieved properties. This brings the question for this study - do the current **operational products** agree with each other under such challenging retrieval conditions?

1.  It looks to me that the saline dust is more scattering than regular dust. Could the authors compare AAI or AAOD products from TROPOMI to verify this?

AR: The EMIT mineralogical map (Fig.1 in the paper) shows Aralkum has negligible iron oxide content compared to sandy deserts (Karakum and Taklamakan), so we expect that the saline dust is less absorbing than the typical desert dust assumed in satellite algorithms. AAI is not a reliable measure of aerosol absorption because it depends on multiple factors (i.e., AOD, SSA, vertical height, surface reflectivity). AAOD is the AOD due to aerosol absorption only, and not an intensive quantity of absorption. AAOD is provided in NASA UVAI products and depends on the aerosol microphysics (size distribution, refractive index) assumed in the algorithm. We looked at the TROPOMI AAOD retrieval, and found that the fresh plume was misclassified as clouds (see below).

[Figure]

2.  Is there any particle size difference between saline dust and regular dust? Perhaps take a look at MODIS AE, although it is not accurate over land, some qualitative clue may appear;

AR: Unfortunately we cannot find PSD information on saline dust from Central Asia. Since AE depends on the aerosol optical models selected during retrieval, and these models are based

on AERONET measurements of Saharan dust, we are not confident about the reliability of AE over land.

3. Do the retrievals of MODIS and VIIRS perform differently over saline desert and other deserts? All the AOD products used in the paper rely on assumptions of aerosol models. The authors did suspect that the uncertainty in these AOD products maybe associated with incorrect model assumptions, but more could be done to verify this. For example, how is the AOD performance here compared to regular deserts, such as North Africa and the Middle East? Could the difference be explained by the different aerosol absorption, vertical distribution or surface reflectance? Some retrieval experiments can even be done, if not too much difficulty, by adjusting the aerosol absorption at least.

AR: Unfortunately, due to lack of AERONET sites or other in situ measurements, we are unable to quantify the AOD performance for the Aralkum saline dust. As far as we know, there were no previous studies on the AOD accuracy for this region, neither. Because of this limitation, the focus of this study is the AOD consistency between **operational products**.

Due to limited time and resources, we are unable to perform extensive algorithm refinements. However, we have added new results for VIIRS EPS AOD by using a dust optical model instead of the selected urban model. We also added new results on MISR retrievals using the research algorithms with better constraint on particle properties.

In short, the authors should provide a comparison between the characteristics and retrieval performance of Aralkum desert and other regular deserts to gain insight into the unique characteristic of saline dust, as well as its satellite retrieval.

Finally, I wonder why the CALIPSO AOCHs from 532 and 1064 nm channels are different?

AR: The CALIOP ALH for 532 and 1064 nm channels have minor differences in this case study. In general, the 532 nm channel is more susceptible to signal loss in the presence of dense absorbing aerosols (e.g., smoke), whereas the 1064 nm channel is able to penetrate deeper into the aerosol layer (Torres et al. 2013).

Reviewer #3:

This paper presents a satellite dust retrieval intercomparison study over Central Asia, with respect to a series of dust storms produced by the Aralkum Desert in May 2018. The authors analyze data from multiple satellite instruments (OMPS, TROPOMI, EPIC, MODIS, VIIRS, CALIOP) and retrieval algorithms over both land and sea so as to understand the performance of the retrievals and the locations where particular caution is required in interpreting the retrieval output. In particular biases are apparent over the turbid waters of the Caspian Sea. Overall I find this paper to be acceptable and straightforward, providing a modestly informative intercomparison of UV and visible satellite dust retrieval products over the Central Asian region.

I think the biggest gap in this paper concerns the apparently saline nature of the dust storms being analyzed. The word saline appears very prominently in the title of the paper, so it is a bit disappointing that this forms such a minor aspect in the analysis and discussion. I see that this is discussed in a couple of paragraphs in the Conclusions, but actually the analysis in the paper is about the performance of the dust retrievals more broadly. The dust is assumed to be composed of a more general desert dust type. It therefore seems slightly misleading to state that this paper is about saline dust.

AR: The main objective of this paper is to investigate the inconsistency and potential biases of **operational satellite aerosol products** in characterizing the saline dust from Aralkum, which is likely insufficiently represented by the candidate aerosol optical models considered in satellite algorithms. As discussed in the Introduction section, in situ aerosol measurements (e.g., AERONET) near Aralkum are nonexistent, and thus no data are available to either constrain the aerosol microphysics assumptions or validate the retrieved aerosol properties. This brings the question that motivates this study - do the current products agree with each other for such challenging retrieval conditions?

Due to limited time and resources, we are unable to perform extensive algorithm refinements. However, we have added new results for VIIRS EPS AOD by using a dust optical model instead of the selected urban model. We also added new results on MISR retrievals using the research algorithms with better constraint on the particle properties.

Moreover, I find it disappointing how few of the retrievals analyzed are over the Aralkum itself. The Aralkum has a relatively dynamic surface environment, with the area of the remaining lake varying both interannually and seasonally, and so I would expect this to be a challenging area for satellite dust retrievals that would be worth a more focused analysis. A related point is made by the authors themselves in the Conclusions around line 580 with respect to UVAI retrievals over the Caspian Sea. However most of the analysis of dust activity is performed over Central Asia more generally, and over the Caspian. This seems like a missed opportunity. Given that the paper is prominently declared to be about saline dust from the Aralkum, I feel that this subject would warrant a deeper analysis of retrievals in the direct vicinity of the Aralkum itself, such an analysis could be an extra subsection or even a whole section within the paper.

AR: The analyzed scenes do include retrievals from Aralkum and its adjacent regions - we just did not single them out, because there are very few retrievals in the direct vicinity, either due to the heavy dust being misclassified as clouds or the marginal QA (which are excluded from our analysis). The low QA reflects the large residuals in the spectral fitting due to AOD saturation and/or inaccurate aerosol optical models and surface reflectances. The reviewer is correct about the dynamic surfaces (e.g. land/water mask) of the Aral Sea region which are challenging for satellite retrievals. In my opinion, it would require multiple case studies or more extensive analysis to look at the effects of dynamic surface changes, e.g., during dry and wet periods.

**Specific comments**

Line 1: "saline dust emission from…". I don't think it is necessarily the emission that is being characterized, rather it is the aerosol above the Aralkum. Perhaps fairer to say "saline dust aerosol emitted from…"

AR: Fixed.

Line 51: I would say that dust activity over the Sahara Desert has also been frequently analyzed over the last few decades.

AR: Correct. Sahara and other major deserts have been studied more extensively.

Line 350: it is worth mentioning here that the SEVIRI Dust RGB composite image is in the infrared, an important distinction with respect to the rest of the (UV and visible) data sources used in this paper.

AR: We have added new sections on thermal infrared-based techniques (algorithm descriptions in Section 2.3 and results in Section 3.4), and described SEVIRI Dust RGB composite as a useful qualitative dust detection product in Central Asia.

Figure 3(g): I think it would be worth adding a vertical line (maybe dashed) to mark the UVAI threshold of 2 that is discussed in the paragraph starting on line 386.

AR: We added thicker vertical gridlines to Fig. 3(g) to help readers see the thresholds.

Line 429: surely the wide geographical range (34-50°N, 47-70°E) refers to the Central Asian region, not the Aral basin.

AR: Correct. This domain covers the Aralkum dust plume and areas not affected by the plume, so we have sufficiently large sample sizes with both clean and dusty retrievals from each algorithm.

Figure 5: the map panels are a little small here. I recommend transposing the panels into a more vertical configuration (i.e. 5 rows x 3 columns), this would make the maps bigger.

AR: We have adjusted the map domain size to improve legibility. We will make further changes if needed.

Reviewer #4:

Not being a reviewer, I will only make a short comment and not a full review.

As it often happens, the thermal infrared is completely absent from this paper. I can understand if the authors do not wish to include it, but then it should be more clear from the title that the study focuses only on UV and VIS observations. In section 2, being entitled "Review of Aerosol

Remote Sensing Over Desert Areas", the thermal infrared should at least be mentioned, even if then the authors state that they will not use it.

For the record, the event analysed in this paper is clearly seen with IASI.

AR: Thanks for the suggestions. We have added new results (Section 3.4) to compare four thermal-infrared AOD products derived from IASI using LMD, ULB, MAPIR, and IMARS algorithms.

References:

Sayer, A. M., Hsu, N. C., Bettenhausen, C., Ahmad, Z., Holben, B. N., Smirnov, A., ... & Zhang, J. (2012). SeaWiFS Ocean Aerosol Retrieval (SOAR): Algorithm, validation, and comparison with other data sets. *Journal of Geophysical Research: Atmospheres*, *117*(D3).

Sayer, A. M., Hsu, N. C., Lee, J., Bettenhausen, C., Kim, W. V., & Smirnov, A. J. J. O. G. R. A. (2018). Satellite Ocean Aerosol Retrieval (SOAR) algorithm extension to S‑NPP VIIRS as part of the "Deep Blue" aerosol project. *Journal of Geophysical Research: Atmospheres*, *123*(1), 380-400.

Torres, O., Ahn, C., and Chen, Z.: Improvements to the OMI near-UV aerosol algorithm using A-train CALIOP and AIRS observations, Atmospheric Measurement Techniques, 6, 3257–3270, https://doi.org/10.5194/amt-6-3257-2013, 2013.

---

## Referee Report (RR1)

***Referee comment on*** **"Analysis of a saline dust storm from the Aralkum Desert – Part 1: Consistency between multisensor satellite aerosol products" by Xin Xi et al.**

The authors' revisions to the paper are substantial and worthwhile, adding to the analysis MISR multiangle visible AOD retrievals and multiple infrared AOD retrievals from IASI, as well as dust aerosol height retrievals from both MISR and IASI. It is now an extensive multisensor inter-comparison study with its focus on the Central Asian region, which makes for a more compelling paper overall. An advantage of including the MISR retrievals is that alongside the AOD the MISR retrieval algorithm also estimates the non-spherical dust contribution, the fine mode fraction, and the single scattering albedo, all of which provide additional insights when considering dust activity in the vicinity of the Aralkum where it is to be expected that saline dust may contribute to the overall dust burden. In general it seems to be a well-argued response by the authors to the reviewers' and the commenter's points.

The MISR AOD analysis in the new Figure 9 seems to me to be a substantial extra contribution in this updated manuscript version. Figure 9 addresses very effectively my main critiques of the previous version as to an apparent shortage of investigations of the aerosol type (i.e. saline dust) and of dust retrievals in the immediate vicinity of the Aralkum. In particular it is very interesting to see the performance of the Research Aerosol algorithms with respect to the operational retrieval output that is more widely used.

The addition of the IASI AOD and altitude retrievals, as well as the MISR altitudes, is also a positive contribution to this updated manuscript. Including infrared retrievals highlights further the wider range of available satellite dust aerosol retrieval algorithms, and in Figure 14 these appear to display potential advantages over the Aralkum region.

As a more specific comment/question to Figure 9, it looks to me as though the RA retrievals are being performed over some of the water bodies in the vicinity of the Aral Sea? Comparing with the satellite image in Figure 19 it looks as though there are RA retrievals over the remaining waters of the South Aral Sea (west and east) and over the lake on the border between Uzbekistan and Turkmenistan. The main areas without retrievals appear to be the areas of the Aral Sea lakebed in the immediate vicinity of the remaining lake. In any case, the RA retrievals appear to be a clear improvement over the operational product in this region.

---

## Author Response (AR2)

Referee comment on "Analysis of a saline dust storm from the Aralkum
Desert – Part 1: Consistency between multisensor satellite aerosol products"
by Xin Xi et al.

The authors' revisions to the paper are substantial and worthwhile, adding to the analysis MISR multiangle visible AOD retrievals and multiple infrared AOD retrievals from IASI, as well as dust aerosol height retrievals from both MISR and IASI. It is now an extensive multisensor inter-comparison study with its focus on the Central Asian region, which makes for a more compelling paper overall. An advantage of including the MISR retrievals is that alongside the AOD the MISR retrieval algorithm also estimates the non-spherical dust contribution, the fine mode fraction, and the single scattering albedo, all of which provide additional insights when considering dust activity in the vicinity of the Aralkum where it is to be expected that saline dust may contribute to the overall dust burden. In general it seems to be a well-argued response by the authors to the reviewers' and the commenter's points.

The MISR AOD analysis in the new Figure 9 seems to me to be a substantial extra contribution in this updated manuscript version. Figure 9 addresses very effectively my main critiques of the previous version as to an apparent shortage of investigations of the aerosol type (i.e. saline dust) and of dust retrievals in the immediate vicinity of the Aralkum. In particular it is very interesting to see the performance of the Research Aerosol algorithms with respect to the operational retrieval output that is more widely used.
The addition of the IASI AOD and altitude retrievals, as well as the MISR altitudes, is also a positive contribution to this updated manuscript. Including infrared retrievals highlights further the wider range of available satellite dust aerosol retrieval algorithms, and in Figure 14 these appear to display potential advantages over the Aralkum region.

As a more specific comment/question to Figure 9, it looks to me as though the RA retrievals are being performed over some of the water bodies in the vicinity of the Aral Sea? Comparing with the satellite image in Figure 19 it looks as though there are RA retrievals over the remaining waters of the South Aral Sea (west and east) and over the lake on the border between Uzbekistan and Turkmenistan. The main areas without retrievals appear to be the areas of the Aral Sea lakebed in the immediate vicinity of the remaining lake. In any case, the RA retrievals appear to be a clear improvement over the operational product in this region.

Response: Thank you for the positive feedback. If I understand the reviewer correctly, the final comment is about the spatial coverage/availability of MISR RA retrievals. Particle properties are retrieved everywhere (same as AOD retrieval), but reported only for large AODs since the information content is low under aerosol loadings. The RA algorithm uses solely MISR NDVI for land/water discrimination, and also relies on MAIAC for the prescribed surface. If there is no prescribed surface from MAIAC we can't run that portion of the algorithm. Additionally, the NDVI for bright salt flats in the immediate vicinity of the lake is very low, which can make land/water discrimination challenging for an instrument with only limited spectral reach (446-865 nm). Because of potential biases in the land/water discrimination, Fig. 9 shows some discontinuity (and biases) of particle property retrievals over the remaining water bodies of Aral Sea and the

Sarygamysh Lake. Another factor is that the AI/ML model is currently trained only over land (using MISR RA geophysical output) but applied everywhere, including water.